# Tuning orbital orientation endows molybdenum disulfide with exceptional alkaline hydrogen evolution capability

Yipeng Zang[1], Shuwen Niu[1], Yishang Wu[1], Xusheng Zheng[2], Jinyan Cai[1], Jian Ye[2], Yufang Xie[1], Yun Liu[1], Jianbin Zhou[1], Junfa Zhu [2], Xiaojing Liu[1], Gongming Wang [1] & Yitai Qian[1]

Molybdenum disulfide is naturally inert for alkaline hydrogen evolution catalysis, due to its unfavorable water adsorption and dissociation feature originated from the unsuitable orbital orientation. Herein, we successfully endow molybdenum disulfide with exceptional alkaline hydrogen evolution capability by carbon-induced orbital modulation. The prepared carbon doped molybdenum disulfide displays an unprecedented overpotential of 45 mV at 10 mA cm$^{-2}$, which is substantially lower than 228 mV of the molybdenum disulfide and also represents the best alkaline hydrogen evolution catalytic activity among the ever-reported molybdenum disulfide catalysts. Fine structural analysis indicates the electronic and coordination structures of molybdenum disulfide have been significantly changed with carbon incorporation. Moreover, theoretical calculation further reveals carbon doping could create empty 2p orbitals perpendicular to the basal plane, enabling energetically favorable water adsorption and dissociation. The concept of orbital modulation could offer a unique approach for the rational design of hydrogen evolution catalysts and beyond.

[1] Hefei National Laboratory for Physical Sciences at the Microscale and Department of Chemistry, University of Science & Technology of China, 230026 Hefei, Anhui, China. [2] National Synchrotron Radiation Laboratory, University of Science & Technology of China, 230029 Hefei, China. These authors contributed equally: Yipeng Zang, Shuwen Niu. Correspondence and requests for materials should be addressed to X.L. (email: liuxj206@ustc.edu.cn) or to G.W. (email: wanggm@ustc.edu.cn)

Sustainable hydrogen production in an economical way is the key to building new hydrogen-based energy system[1]. Water electrolysis driven by renewable energy sources such as solar, wind, or geothermal energy has been regarded as the most promising way for sustainable hydrogen production[2,3]. Despite the fact that water electrolysis is first discovered in acidic condition, alkaline electrolysis is more preferred in industrial plants, due to the robustness of electrode materials, long lifetime, and cheap electrolyzer construction[4]. Electrocatalysts, as the heart of the electrolyzers, significantly affect the cell efficiency[5]. Unfortunately, precious platinum-based materials till now are still the state-of-the-art hydrogen evolution reaction (HER) catalysts in both acidic and alkaline conditions[6,7]. In this regard, developing non-noble metal-based HER catalysts to replace Pt has been one of the focal points over the past decades[8–14].

Molybdenum disulfide ($MoS_2$), a typical cost-effective layered transition metal dichalcogenide[15–18], has been proved both theoretically and experimentally to be highly active for HER catalysis in acidic condition[16–26]. The edged Mo and S atoms with a free energy of H adsorption close to zero are commonly believed to be the catalytic sites[19,27–30]. Unfortunately, $MoS_2$ is found to be inactive in alkaline condition[7,31,32], while the alkaline catalysis is more practically viable. The inert alkaline electrolysis is mainly attributed to the sluggish water adsorption and dissociation dynamics, which is, in essence, determined by the intrinsic structural feature of $MoS_2$[19,33–35]. As the dominated orbital compositions of the conduction band in $MoS_2$, both the Mo $4d_{z^2}$ orbitals in the central sublayer and the S $3p_{x,y}$ orbitals in the outermost sublayers possess unfavorable charge interaction with water molecule due to the steric effect ($d_{z^2}$) and unsuitable orbital orientation ($3p_{x,y}$) in the basal plane, which essentially hinder the water adsorption and dissociation on $MoS_2$ (Fig. 1). Although improving the conductivity of $MoS_2$[31,36,37] or surface modification with water adsorption components[19,33,34] has shown enhanced alkaline electrolysis, the overall performance of $MoS_2$ is still far from being satisfactory for practical alkaline electrolysis and studies on this issue are also very limited. Moreover, the understanding of the alkaline catalytic essences especially at atomic levels is also absent. Therefore, rationally tailoring the catalytic sites to endow $MoS_2$ with efficient alkaline HER catalytic activity is highly desirable but extremely challenging as well.

To essentially manipulate the intrinsic properties of $MoS_2$ for alkaline HER catalysis, tuning the orbital orientation of the $MoS_2$ layer to promote water adsorption and dissociation dynamics is the key. Considering carbon with smaller atomic radius and less

electrons owns more empty and lower-energy 2p valence orbitals than sulfur, partially substituting S with carbon in $MoS_2$ could generate $sp^2$ hybrid sites in the outermost sublayers of $MoS_2$, which consequently vacate one p orbital perpendicular to the basal plane for water adsorption and activation. Herein, we successfully endow $MoS_2$ nanosheets with exceptional alkaline HER activities by carbon-induced surface orbital orientation modulation. Carbon is in-situ incorporated into $MoS_2$ through a unique incomplete sulfurization of $Mo_2C$. The prepared carbon doped $MoS_2$ ($C–MoS_2$) nanosheets display an unprecedented overpotential of 45 mV at 10 mA cm$^{-2}$ in alkaline condition, which represents the best alkaline HER activity among the ever-reported $MoS_2$-based catalysts. X-ray photoelectron spectroscopy (XPS) and X-ray absorption spectroscopy (XAS) systematically reveal the structural and electronic evolution of $MoS_2$ after carbon doping. Moreover, density functional theory (DFT) analysis indicates carbon doping can generate empty p orbitals perpendicular to the basal plane of $MoS_2$ for water adsorption and dissociation, which is essential for the alkaline HER catalysis. More importantly, the capability to endow materials with the properties which are not readily available in nature by rational orbital modulation offers a new vision for the design of HER catalysts and beyond.

## Results

**Synthesis and structural characterization of C–MoS$_2$.** C–MoS$_2$ and $MoS_2$ were synthesized via a controlled sulfurization of $Mo_2C$, which is illustrated in Fig. 2a. $Mo_2C$ as the precursor was obtained by a previously developed method (details, see Methods)[38]. Then, the as-synthesized $Mo_2C$ was controllably sulfurized to C–MoS$_2$ and $MoS_2$ in a home-built tube furnace with sulfur powder as the sulfur source and argon as the carrier gas (details, see Methods). The synthesized $Mo_2C$, C–MoS$_2$, and $MoS_2$ were further characterized by field-emission scanning electron microscopy (SEM), X-ray diffraction (XRD), Raman spectrum, and transmission electron microscopy (TEM), respectively. SEM images in Supplementary Fig. 1 indicate the carbon fibers of the carbon cloth (CC) substrate are uniformly coated with porous $Mo_2C$ thin film, which becomes rougher after sulfurization treatment. XRD patterns (Fig. 2b) clearly reveal the $Mo_2C$ (JCPDS No. 00-035-0787) can be well converted to hexagonal $MoS_2$ (JCPDS No. 01-073-1508) after controlled sulfurization treatment. The sharp diffraction feature of $MoS_2$ with deep sulfurization suggests improved crystallinity. In addition, the sulfurization process can also be revealed by Raman spectroscopy (Supplementary Fig. 2). The fingerprint bands of $Mo_2C$ located at 662.5, 816.4, and 991.8 cm$^{-1}$ disappear after sulfurization treatment[39]; meanwhile, new bands centered at around 376 and 402 cm$^{-1}$ emerge, which can be assigned to the in-plane ($E_{2g}$) and out-plane ($A_{1g}$) Mo–S phonon mode vibration of $MoS_2$, respectively[40]. Moreover, the wavenumbers of $E_{2g}$ and $A_{1g}$ of C–MoS$_2$ display a positive shift relative to the $MoS_2$, probably due to the susceptibility of electron–phonon coupling induced by carbon doping[41,42]. Taken together, all these results illustrate sulfurization treatment can well control the conversion from $Mo_2C$ to $MoS_2$.

Furthermore, TEM was employed to acquire the microstructural features of $Mo_2C$, C–MoS$_2$, and $MoS_2$, as shown in Supplementary Fig. 3 and Fig. 2c. The high-resolution TEM (HRTEM) images exhibit well-resolved lattice fringes, where the interplanar spacing of 0.23 nm is assigned to the (101) plane of $Mo_2C$ and 0.62 nm is assigned to the (002) plane of $MoS_2$, respectively. Interestingly, both C–MoS$_2$ and $MoS_2$ prepared by sulfurization treatment are composed of ultrathin layered $MoS_2$ nanosheets. It is also worth mentioning that no obvious carbon coating is observed on the surfaces of $Mo_2C$, C–MoS$_2$, and

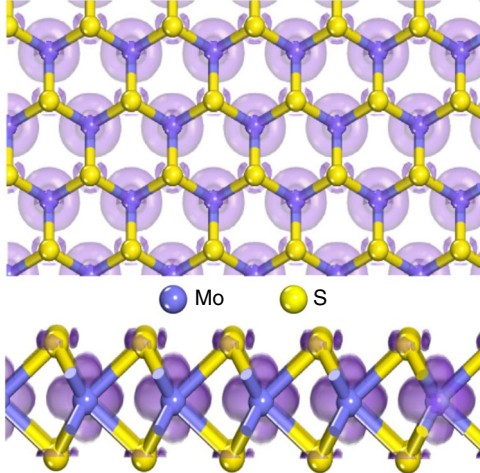

**Fig. 1** The orbital analysis of the $MoS_2$. The top-view (upper) and side-view (lower) orbital compositions of the conduction band in $MoS_2$

Mo    S

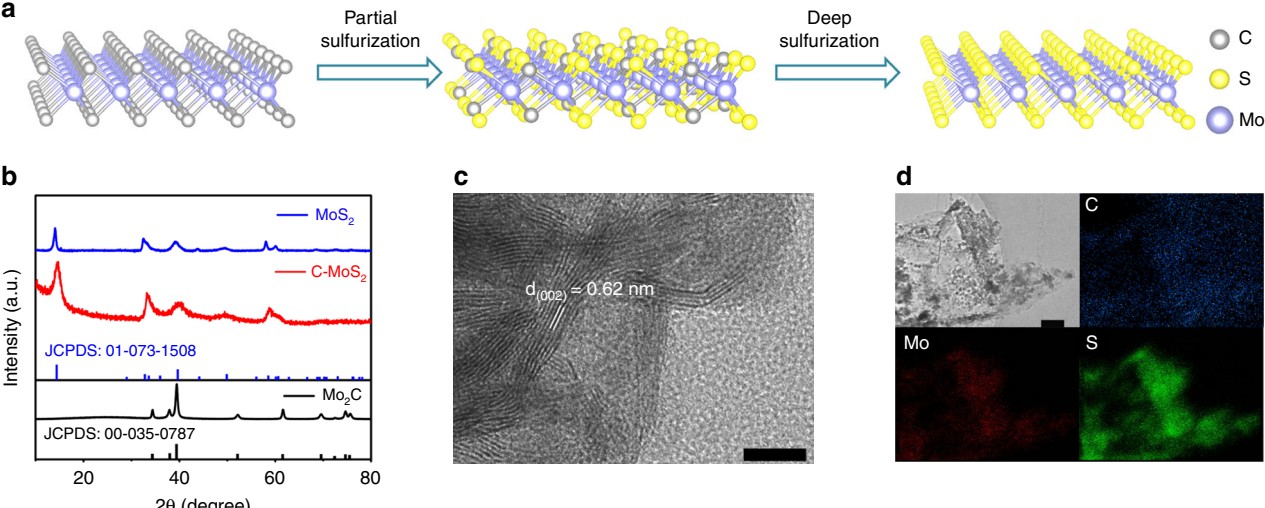

**Fig. 2** Synthesis and characterization of C–MoS₂ nanosheets. **a** The schematic illustration of the synthesis of C–MoS₂ and MoS₂. **b** The XRD patterns of Mo₂C, C–MoS₂, and MoS₂. **c** The HRTEM image of C–MoS₂. The scale bar is 10 nm. **d** The TEM image and the corresponding element mapping images of C, Mo, and S elements in C–MoS₂. The scale bar is 200 nm

MoS₂ (Supplementary Fig. 2 and Fig. 2c), offering an ideal platform to probe the intrinsic effects of carbon dopants on the properties of MoS₂. In addition, the TEM image and the corresponding energy-dispersive X-ray spectroscopy (EDX) mapping images of C–MoS₂ reveal homogeneous elemental distribution of C, Mo, and S elements in C–MoS₂ (Fig. 2d), also suggesting possible carbon doping in the MoS₂.

**Chemical states and coordination structures of C–MoS₂.** Given that EDX spectroscopy only provides composition information that is not convincing enough to prove carbon doping into MoS₂, XPS, and XAS are further used to probe localized electronic states and coordination structures of C–MoS₂. Figure 3a shows the XPS Mo 3d spectra of Mo₂C, MoS₂, and C–MoS₂, respectively. Obviously, the typical characteristics of Mo–C bonds are observed at 231.1 and 228.0 eV in Mo₂C, while a pair of new peaks located at 232.6 and 229.5 eV appear after deep sulfurization, which can be assigned to the Mo $3d_{3/2}$ and Mo $3d_{5/2}$ of Mo–S bond in MoS₂, respectively[43]. Strikingly, the Mo 3d profiles in C–MoS₂ exhibit obvious shoulders at the lower binding energy region, suggesting the existence of multiple chemical states of Mo. The Mo 3d spectrum of C–MoS₂ can be further deconvoluted into two pairs of peaks corresponding to the chemical states of S–Mo–S and S–Mo–C, which indicates carbon has been substitutionally doped into the MoS₂ lattices (The fitting parameters are summarized in Supplementary Table 1). Meanwhile, the C 1s spectra of C–MoS₂ and Mo₂C (Supplementary Fig. 4, the fitting parameters in Supplementary Table 2) also support the existence of Mo–C bonds in C–MoS₂. In addition, the slight shift of S 2 s position suggests the chemical environments around S atoms also slightly changes after carbon doping, which could be originated from the electronegativity difference between C and S.

To probe the evolution of localized coordination environment in MoS₂ induced by carbon doping, X-ray absorption near-edge structure (XANES) and extended X-ray absorption fine structure (EXAFS) are performed. Figure 3b shows the Mo K-edge XANES spectra of Mo₂C, C–MoS₂, MoS₂, and Mo foil, in which Mo foil is used as the reference. Apparently, the Mo foil possesses the smallest energy of absorption edge. Moreover, the enlarged Mo K-edge XANES curves (inset, Fig. 3b) reveal that the near-edge absorption energy of C–MoS₂ located between MoS₂ and Mo₂C indicates the average electron density around Mo in C–MoS₂ is

higher than MoS₂ but lower than Mo₂C, which is consistent to the XPS Mo 3d spectra and also suggests the existence of Mo–C bonds in C–MoS₂. In addition, Mo K-edge extended XAFS oscillation functions $k^2\chi(k)$ of MoS₂, C–MoS₂, Mo₂C, and MoS₂ mixed with small amounts of Mo₂C (denoted as Mo₂C + MoS₂) are presented in Supplementary Fig. 5a, respectively. Clearly, the oscillation of C–MoS₂ is very similar to MoS₂, but quite different from Mo₂C and Mo₂C + MoS₂, convincingly indicating the C–MoS₂ possesses similar structure to MoS₂ and there is no obvious Mo₂C remaining in C–MoS₂. Moreover, the XPS depth profiling analysis also does not show obvious existence of Mo₂C (Supplementary Fig. 5b). Figure 3c displays the corresponding R-space curves of Mo after $k^2$ [χ(k)] weighted Fourier transform. The shoulder peaks at around 1.9 and 2.9 Å in MoS₂ are originated from the Mo–S and Mo–Mo vectors, respectively[44]. Considering the positions of Mo–C and Mo–O are very close and no obvious Mo–O states are observed in the XPS Mo 3d spectra, we reasonably believe the peak at 1.3 Å in C–MoS₂ could be assigned to the Mo–C bond[45]. To more clearly reveal the coordination structures in C–MoS₂, wavelet transform (WT) with high resolution in both k and R spaces of Mo K-edge EXAFS oscillation are further performed. The whole WT contour spectra are shown in the Fig. 3d (the upper). The intensity maxima at R = 1.9 and 2.9 Å are attributed to the Mo–S and Mo–Mo bonds, respectively. Owing to the small bond length difference between Mo–C and Mo–S, more-refined first shell analysis of MoS₂ and C–MoS₂ is further carried out (Fig. 3d, the lower). Apparently, MoS₂ possesses only one contour centered at about $k = 5.4 \text{ Å}^{-1}$, originated from the Mo–S bond, while C–MoS₂ presents one more maximum intensity at $k = 4.6 \text{ Å}^{-1}$, which is very close to the Mo–C in Mo₂C centered at about $k = 4.5 \text{ Å}^{-1}$ (Supplementary Fig. 6a) and far from the Mo–O in MoO₂ with the maximum intensity at $k = 6.0 \text{ Å}^{-1}$ (Supplementary Fig. 6b), and thus can be well assigned to the Mo–C. Meanwhile, we also conduct the EXAFS fitting in the range of 1.1 to 2.3 Å by using the Mo–O and Mo–C path at 1.3 Å, respectively (Supplementary Fig. 6c,d). Based on the fitting parameters presented in the Supplementary Table 3, the $R_f$ value for the Mo–O model (2.9%) is much larger than that of the Mo–C path (1.1%), further suggesting the Mo–C path is more likely than the Mo–O path in the C–MoS₂. In light of the formation of Mo–C in C–MoS₂, the coordination environment of MoS₂ may also alter with carbon doping. Furthermore, the EXAFS

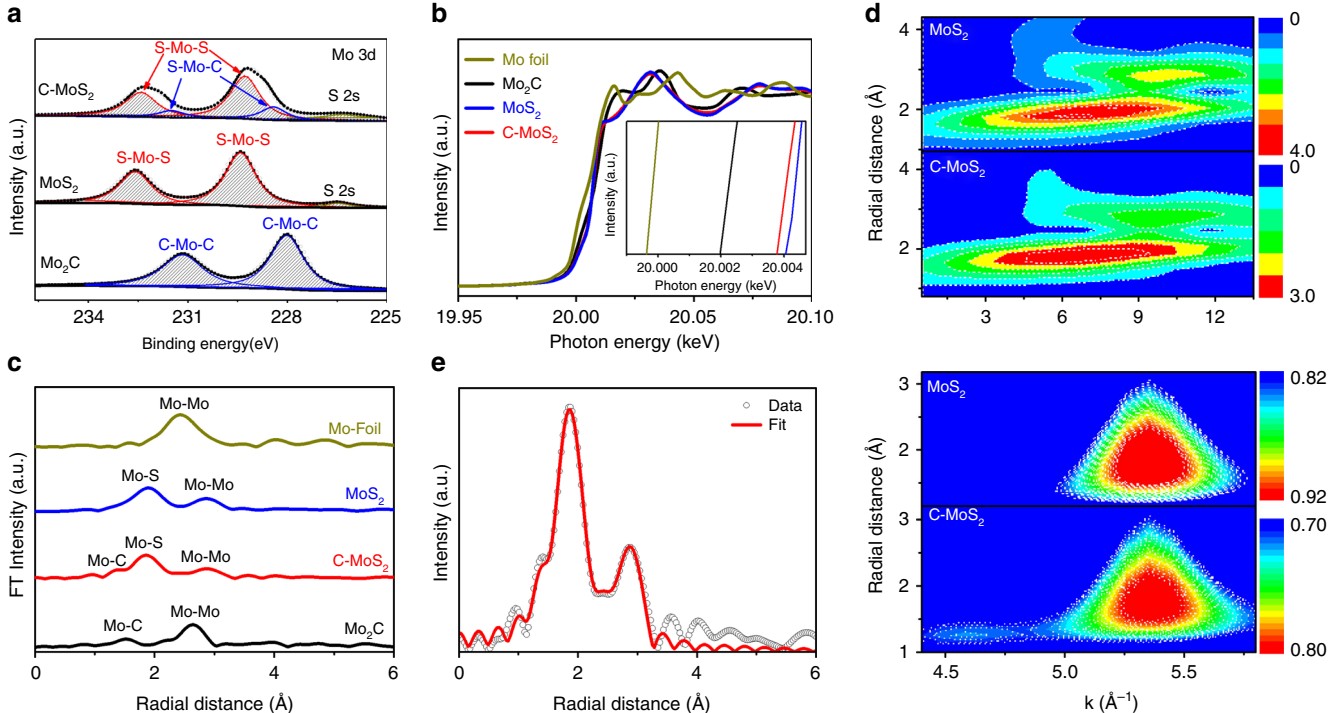

**Fig. 3** Chemical states and coordination structures of C–MoS$_2$. **a** XPS core-level Mo 3d spectra of Mo$_2$C, MoS$_2$, and C–MoS$_2$, respectively. **b** The normalized Mo K-edge XANES spectra. Inset: the enlarged Mo K-edge XANES spectra. **c** Fourier transform (FT) of Mo K-edge of Mo$_2$C, C–MoS$_2$, MoS$_2$, and Mo foil. **d** The whole contour plots of wavelet transform (WT) of MoS$_2$ and C–MoS$_2$ (upper) and the more-refined first shell analysis (lower). **e** Radial distribution of Fourier-transformed EXAFS signal of C–MoS$_2$

data of MoS$_2$ and C–MoS$_2$ are fitted through the Artemis to get detailed structural parameters (Supplementary Fig. 7 and Fig. 3e). The coordination number of Mo–S is fitted to be around 6 for the MoS$_2$, while the coordination number of Mo–S is deceased to 4.8 and the coordination number of Mo–C is 1.1 in the C–MoS$_2$. The detailed fitting parameters are summarized in Supplementary Table 4. Although we cannot exclude the existence of Mo$_2$C clusters in C–MoS$_2$ that is beyond the detection limits of the currently used characterization techniques, our experimental results clearly reveal that carbon is most probably doped into the lattices of MoS$_2$ in the form of Mo–C bond.

**Alkaline hydrogen evolution catalysis of C–MoS$_2$.** The alkaline HER catalysis is evaluated in 1.0 M KOH solution using a typical three-electrode system with the studied materials as the working electrodes, Hg/HgO as the reference electrode and graphite rod as the counter electrode. Figure 4a shows the linear sweep voltammetry (LSV) curves of CC, Mo$_2$C, C–MoS$_2$, MoS$_2$, and Pt/C with a scan rate of 5 mV s$^{-1}$, in which the blank CC and benchmark Pt/C are used as control samples. Impressively, C–MoS$_2$ delivers higher current density and lower current onset potential than CC, MoS$_2$, and Mo$_2$C, indicating carbon doping can substantially improve the catalytic performance toward HER in alkaline condition. Carbon doping enables the overpotential of C–MoS$_2$ at 10 mA cm$^{-2}$ to be as low as 45 mV, which is much better than the 200 mV of Mo$_2$C and 228 mV of MoS$_2$ and even close to the catalytic activity of Pt/C (30 mV). Although the performance of C–MoS$_2$ is not the best among all kinds of the HER catalysts including non-precious and precious catalysts, C–MoS$_2$ represents the best alkaline HER activity among the ever-reported MoS$_2$ (Fig. 4b and Supplementary Table 5 and 6)[7,19,32–34,36,37,46–48], which could provide valuable insights for the active site engineering. The faradaic efficiency of the

C–MoS$_2$ for HER catalysis is estimated to be around 97%. Given that temperature can well control the sulfurization degrees, we also study the sulfurization degree-dependent HER activities (Supplementary Fig. 8). With the increase of the sulfurization temperatures, the HER overpotentials display an inverse volcano-shaped feature. At the temperature below 500 °C, Mo$_2$C cannot be completely converted to MoS$_2$ and the corresponding catalytic activity is far less than that of the C–MoS$_2$, further revealing C doping probably plays vital role in the HER catalysis. To probe the effects of carbon contents, we further study the carbon content dependent HER activities of C–MoS$_2$ (Supplementary Fig. 9). Obviously, with the increase of carbon concentration, the catalytic activity of C–MoS$_2$ increases and reaches a maximum value. However, further increasing the C ratio will result in performance decay probably due to too much destroy of the layered structures of MoS$_2$, suggesting moderate carbon doping in MoS$_2$ is essential for the HER catalysis. Additionally, Tafel slope can be used to probe the effects of carbon dopants on the rate-determining steps during HER process. Figure 4c displays the corresponding Tafel curves. The derived Tafel slope of MoS$_2$ is around 129 mV dec$^{-1}$, suggesting the alkaline HER catalysis on MoS$_2$ undergoes Volmer mechanism and water dissociation is the rate-determining step. Importantly, C–MoS$_2$ exhibits a substantially decreased Tafel slope of 46 mV dec$^{-1}$, clearly revealing the sluggish water dissociation behavior has been significantly improved after carbon doping. Meanwhile, the exchange current densities for Mo$_2$C, MoS$_2$, C–MoS$_2$, and Pt/C derived by Tafel extrapolation are 0.12, 0.25, 0.87, and 1.28 mA cm$^{-2}$, respectively. Clearly, although the exchange current density of C–MoS$_2$ is a little smaller than that of Pt, it is still substantially larger than that of MoS$_2$, also suggesting carbon doping can change the intrinsic catalytic activities of MoS$_2$ for alkaline HER catalysis.

Considering surface area may also affect the HER catalytic performance, we further estimate the electrochemical surface

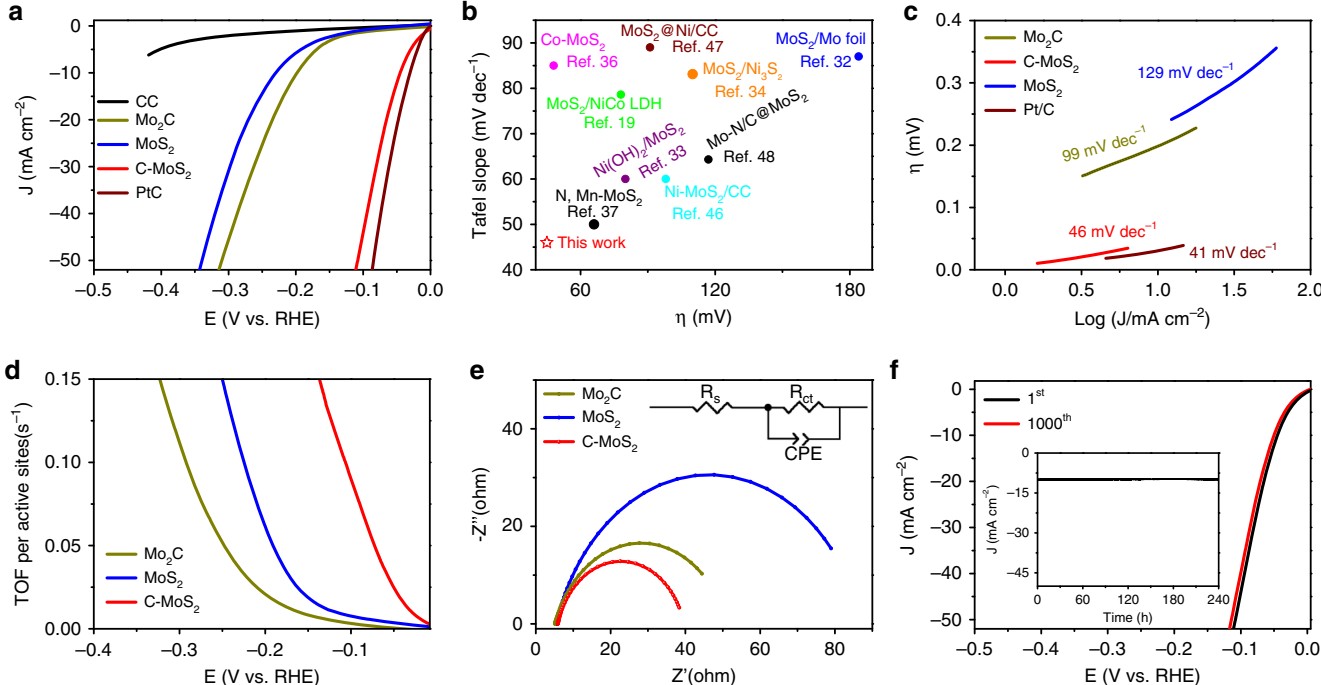

**Fig. 4** Alkaline HER catalysis of C–MoS$_2$. **a** The LSV curves of CC, Mo$_2$C, C–MoS$_2$, MoS$_2$, and Pt/C with IR correction. **b** Performance comparison of C–MoS$_2$ with the ever-reported MoS$_2$-based catalysts in alkaline condition. **c** The corresponding Tafel slopes. **d** The potential-dependent TOF curves of Mo$_2$C, MoS$_2$, and C–MoS$_2$. **e** Nyquist plots of Mo$_2$C, MoS$_2$, and C–MoS$_2$ collected at the potential of 100 mV vs. RHE. **f** The initial and 1000th polarization curves of C–MoS$_2$. The inset is the chronoamperometric curve recorded at −0.1 V vs. RHE without IR correction for a continuous 240 h

areas of the studied materials by deriving the electrochemical double layer capacitance (C$_{dl}$) from the cyclic voltammetry studies (Supplementary Fig. 10)[11]. The C–MoS$_2$ is found to have a larger C$_{dl}$ of 54 mF cm$^{-2}$ than MoS$_2$ (19 mF cm$^{-2}$) and Mo$_2$C (27 mF cm$^{-2}$), indicating carbon doping treatment can also increase the surface area of C–MoS$_2$. To eliminate the influence of surface area and reveal the intrinsic catalytic behavior, turnover frequency (TOF), a surface area independent figure of merit, is calculated to gain the intrinsic per-site activity, as shown in Fig. 4d. Impressively, the TOF values of C–MoS$_2$ are significantly larger than those of MoS$_2$ and Mo$_2$C, clearly revealing carbon doping can essentially promote the per-site activity for HER catalysis. Additionally, electrochemical impedance spectroscopy (EIS) in Fig. 4e presents that C–MoS$_2$ possesses a smaller charge transfer resistance (R$_{ct}$) of 32.1 Ω than MoS$_2$ (83.7 Ω) and Mo$_2$C (59.7 Ω), further suggesting carbon doping in MoS$_2$ can also substantially boost the interfacial electron-transfer kinetics. Besides carbon doping, the superior electrochemical behavior of C–MoS$_2$ is also attributed to the unique carbon doping strategy without involving carbon deposition on the catalyst surface. We also use a conventional post-doping method to introduce carbon and study their electrochemical performance, shown in Supplementary Fig. 11. TEM image clearly illustrates obvious carbon shell deposited on the surface of MoS$_2$, which results in decreased catalytic performance probably due to the blocked active sites for catalysis (Supplementary Fig. 12). Given that the MoS$_2$ is derived from Mo$_2$C which may have tiny amounts of carbon remaining, MoS$_2$ with minimum carbon interference is also prepared by thermal sulfurization of ammonium molybdate (Supplementary Fig. 13). Electrochemical studies reveal that the HER performance of the newly prepared MoS$_2$ is far less than that of C–MoS$_2$, also suggesting carbon dopants play vital roles in the HER catalysis of C–MoS$_2$. Finally, the stability of C–MoS$_2$ is evaluated by polarization cycling and chronoamperometric test, both of which display superior catalytic stability for alkaline electrolysis (Fig. 4f).

XPS and XAFS are further employed to evaluate the C–MoS$_2$ after the stability test (Supplementary Fig. 14). Impressively, there is no big difference in chemical and bonding states in C–MoS$_2$ before and after the durability test, suggesting the robustness of the C–MoS$_2$ for HER catalysis.

## Discussion

To decipher the modulation essence of the carbon dopants in C–MoS$_2$ for alkaline HER catalysis at the atomic level, DFT calculations are further carried out. The C content used in the C–MoS$_2$ model is based on the area ratio of Mo–C and Mo–S states in the XPS Mo 3d of C–MoS$_2$. In light of the charge balance after carbon doping, we also introduce sulfur vacancies (V$_s$) in the structural models. To equally distribute the positions of the C dopants and S vacancies, we considered seven different configurations, as shown in Supplementary Fig. 15. Supplementary Fig. 16 shows one of the typical structures of C–MoS$_2$, where the C dopants are pulled into the sublayer of MoS$_2$, due to the formed stronger Mo–C bond with shorter bond length than the original Mo–S bond. Detailed bond length information is provided in Supplementary Fig. 17. The electron density difference images in Supplementary Figs 18 and 19 clearly reveal that the introduction of carbon can significantly break the unified surface electron distribution of MoS$_2$. More importantly, the orbital analysis in Fig. 5a and b indicates the carbon dopants prefer to form the sp$^2$ hybrid orbitals (highlighted by red dash circle in Fig. 5a), which thus vacates an unhybridized 2p$_z$ orbital perpendicular to the basal plane (highlighted by red dash circle in Fig. 5b). This 2p$_z$ orbital orientation can maximize the head-on orbital overlapping to form sigma bonds, which can potentially offer active sites for water adsorption and activation. This orbital orientation is also consistently observed on the other structural configurations of C–MoS$_2$, as shown in Supplementary Fig. 15. Additionally, the detailed partial density of states (PDOS) analysis

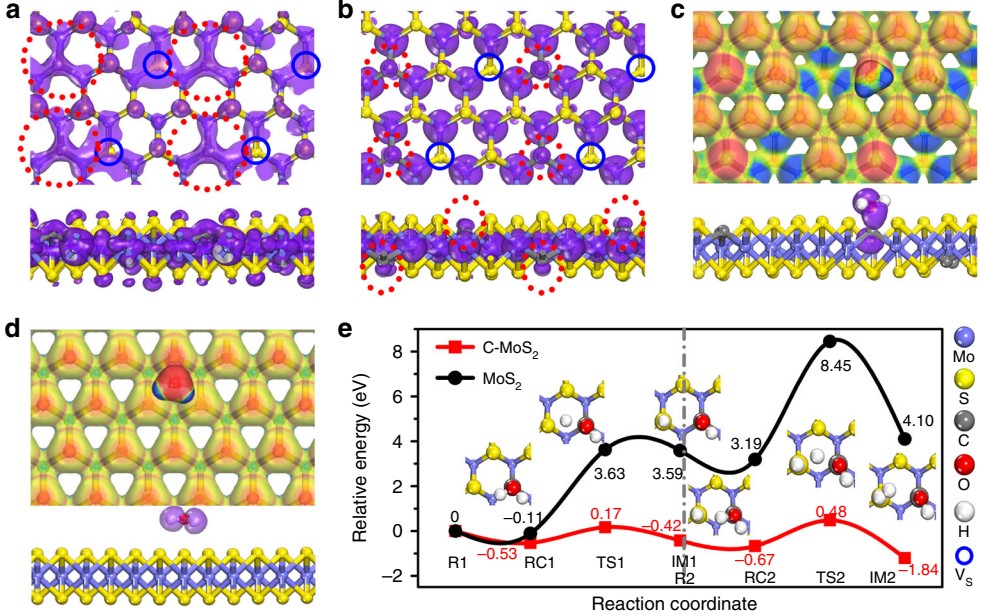

**Fig. 5** The structural analysis and the catalytic pathway of C–MoS$_2$. The top-view and side-view sp$^2$ hybrid orbitals (highlighted by red dash circle) at the top of valence band (**a**) and the empty 2p orbitals (highlighted by red dash circle) perpendicular to the basal plane at the bottom of conduction band (**b**) of C–MoS$_2$. The top-view electrostatic potential of water adsorbed on the basal plane of C–MoS$_2$ (**c**) and MoS$_2$ (**d**) and the corresponding side-view bonding and non-bonding orbitals. **e** The relative energy diagram along the reaction coordinate, including the first (left panel) and second (right panel) water dissociation process on the basal plane of MoS$_2$ and C–MoS$_2$, respectively. R reactant, RC reactant complex, TS transition state, IM intermediate

(Supplementary Fig. 20) indicates that carbon doping can well increase the electrical conductivity of the MoS$_2$ and thus could benefit the electron transportation and enhance the catalytic activity. Although conductivity is important for electrocatalysis, we reasonably believe the exceptional alkaline HER activity of C–MoS$_2$ is mainly stemmed from the C-induced orbital tuning for water adsorption and dissociation, considering 1-T MoS$_2$ with better conductivity does not exhibit such exceptional activity[49]. Moreover, the electrostatic potential mapping images (Fig. 5c and Supplementary Fig. 21) and the electron density difference slices (Supplementary Fig. 22) further reveal superior water adsorption on the basal plane of C–MoS$_2$, which is confirmed by the apparent charge transfer and the formation of remarkable chemical bonding orbital between water and C–MoS$_2$ (Fig. 5c), while no apparent orbital bonding or charge interaction occurs on MoS$_2$ (Fig. 5d). We also consider the MoS$_2$ model with only sulfur vacancies. Structural analysis indicates sulfur vacancies indeed do not have obvious effects on the orbital orientation tuning (Supplementary Fig. 23a). The corresponding water adsorption energy on the defected MoS$_2$ (MoS$_2$-Vs) is around −0.23 eV, which is only slightly higher than that on the MoS$_2$. Meanwhile, we also experimentally prepare MoS$_2$ with sulfur vacancies via a typical hydrogen treatment and study their electrochemical properties. The overall performance of the hydrogen treated MoS$_2$ is still far less than that of C–MoS$_2$ (Supplementary Fig. 23b), suggesting the sulfur vacancy may not be the main factor for the exceptional alkaline HER activity of the C–MoS$_2$. Taken together, all the structural information consistently suggest that carbon doping can effectively manipulate the orbital orientation and create more active sites on the basal plane of MoS$_2$ for water adsorption and activation.

Besides structural information, alkaline HER catalytic pathways on the basal plane of MoS$_2$ and C–MoS$_2$ are further studied. Figure 5e shows the relative energy profiles of water adsorption and dissociation on MoS$_2$ and C–MoS$_2$, respectively. As expected, the basal plane of MoS$_2$ is almost inert for alkaline HER catalysis,

with poor water adsorption and giant water dissociation energy barrier. Impressively, carbon doping enables the HER catalysis to proceed on a much lower potential energy surface. Specifically, C–MoS$_2$ owns more favorable adsorption energies (−0.53 eV for RC1 & −0.25 eV for RC2) and much lower transition state energy barrier (0.70 eV for TS1 & 1.15 eV for TS2). From the water adsorption/dissociation configurations on C–MoS$_2$ (Fig. 5e), H$_2$O is adsorbed on the C site, while the nearby S sites assist water dissociation by attracting the H in water molecule. In addition, we also performed the catalytic pathway of Mo$_2$C (001) (Supplementary Fig. 24). Obviously, the reaction on the Mo$_2$C proceeds on a potential energy surface with much higher energy barrier than that on C–MoS$_2$. Moreover, the generated hydrogen adsorbed intermediate (Sur-H) is so stable that the adsorbed hydrogen needs overcome 1.37 eV for H desorption from the carbon site of the surface, which could potentially impede the HER catalysis. The studies on the catalytic pathways clearly reveal carbon doping could intrinsically boost the water adsorption and dissociation kinetics and thus endow MoS$_2$ with exceptional alkaline catalysis by tuning the orbital orientations on the basal plane.

In summary, we have demonstrated carbon doping can intrinsically endow MoS$_2$ with exceptional alkaline HER catalytic capability by tuning the orbital orientation. In comparison with the inert alkaline HER activity of MoS$_2$, the prepared C–MoS$_2$ achieves an overpotential of 45 mV at 10 mA cm$^{-2}$, which is very close to the commercial Pt/C and also represents the best catalytic activity among the ever-reported MoS$_2$ for alkaline HER catalysis. XPS and XAS analysis reveal that the electronic and coordination structures of MoS$_2$ have been significantly changed after carbon doping. DFT studies further indicate that carbon-induced empty 2p orbitals perpendicular to the basal plane of MoS$_2$ enable energetically favorable water adsorption and dissociation, and thus promote the alkaline HER catalytic kinetics. The capability to intrinsically manipulate the catalytic activities by orbital modulation could offer a powerful platform to rationally design HER catalysts and beyond.

## Methods

**Reagents and chemicals**. Ammonium molybdate ((NH$_4$)$_6$Mo$_7$O$_{24}$·4H$_2$O), Potassium hydroxide, Cetyltrimethyl ammonium bromide (CTAB), Thiourea and Sublimed sulfur were purchased from Shanghai Sinopharm Chemical Reagent Co., Ltd. CC was purchased from Shanghai Hesen Electric Co., Ltd.

**Synthesis of Mo$_2$C**. Typically, 3.708 g ammonium molybdate and 1.638 g CTAB were dissolved in 30 mL and 180 mL deionized water with ultrasonic assistance, respectively. Then, the two solutions were mixed homogeneously by magnetically stirring for 30 min, and further aged for another 24 h to obtain a uniform emulsion. A volume of 15 mL of the as-prepared emulsion was transferred into a 20 mL Teflon-lined stainless steel autoclave with the pretreated CC (CC, 2*3 cm$^2$, pre-treated with nitric acid and deionized water, respectively) as the growth substrate. After keeping the autoclave at 200 °C for 20 h, the CC was taken out, cleaned by deionized water and dried at vacuum oven at 60 °C overnight. Finally, Mo$_2$C was obtained by annealing the prepared samples at 900 °C for 6 h in a home-built tube furnace system under argon atmosphere.

**Synthesis of C–MoS$_2$ and MoS$_2$**. C–MoS$_2$ and MoS$_2$ were prepared by a temperature-controlled sulfurization process. The as-prepared Mo$_2$C was sulfur-ized in a tube furnace system for 2 h with powdered sulfur as the sulfur source and argon as the carrier gas (flowing rate at 20 sccm). The sulfurization temperatures can be tuned from 500 to 900 °C. The optimized C–MoS$_2$ was achieved at 600 °C with a partial sulfurization, while MoS$_2$ was fabricated at 900 °C with a deep sulfurization. As comparison, a post-doping strategy was employed to dope carbon into MoS$_2$. The carbon doped MoS$_2$ by post-doping method (PC–MoS$_2$) was achieved by annealing MoS$_2$ at 750 °C for 0.5 h under acetylene flow. MoS$_2$ with minimum carbon interference was synthesized by sulfurizing ammonium molyb-date with S as the sulfur source and argon as the carrier gas (flowing rate at 20 sccm) for 2.0 h under 900 °C[25].

**Structural characterizations**. XRD measurements were performed on a Philips X'Pert Pro Super diffractometer using Cu Kα radiation (λ = 1.54178 Å). The field-emission scanning (FE-SEM) images were collected using a FEI Sirion-200 SEM, while TEM and HRTEM images were acquired at a JEOL-2010 TEM with an acceleration voltage of 200 kV. Raman spectra were recorded on a Renishaw RM 3000 Micro-Raman system. XPS were performed at the photoemission end-station (BL10B) in the National Synchrotron Radiation Laboratory (NSRL), Hefei. To avoid the carbon interference for the binding energy calibration, all the samples were sprayed with tiny amount of Au as an external standard and the obtained binding energies were calibrated using the Au 4f energy (Au 4f$_{7/2}$, binding energy of 84 eV). The XPS fitting was performed using the XPS PEAK41 software with Shirley background subtraction. The XPS database (NIST X-ray Photoelectron Spectroscopy Database, https://srdata.nist.gov/xps/intro.aspx) was used as a refer-ence to assign the possible chemical states. During fitting, the peak intensity ratio and the full width at half maximum (FWHM) were well constrained in a reasonable range. XAS measurements were conducted at the beamline (BL14W1) of Shanghai National Synchrotron Radiation Facility (SSRF, China). The EXAFS data were disposed according to the standard procedures through the ATHENA module implemented in the IFEFFIT software packages[50]. The quantitative curve-fittings were conducted for the Fourier-transformed k$^2$-weighted χ(k) in the R-space with a Fourier transform k-space range of 3.1–11 Å$^{-1}$ by employing the module ARTE-MIS 4 of IFEFFIT. The backscattering amplitude F(k) and phase shift Φ(k) were calculated by FEFF8.0 code. While the curve-fitting, all the amplitude reduction factor S$_0^2$ was set to the best-fit value of 0.85 determined from fitting the data of MoS$_2$. In order to fit the curves in the R-range of 1.1–3.4 Å, we considered Mo–C, Mo–Mo and Mo–S paths as the central-peripheral. For each path, the structural parameters, like coordination number (N), interatomic distance (R), and edge-energy shift (ΔE$_0$) were opened to be varied. For the Mo–C, Mo–S, and Mo–Mo coordination, the common adjustable parameters of ΔE$_0$ and σ$^2$ were employed to reduce the number of free parameters.

**Electrochemical measurements**. All the electrochemical characterizations were performed in a three-electrode system on CHI 760E electrochemical workstation. The studied materials grown on CC were used as working electrodes, while Hg/HgO electrode and graphite rod were employed as the reference and counter electrodes, respectively. All the measurements were conducted in 1.0 M KOH electrolyte and the potentials vs. Hg/HgO were converted with respect to the reversible hydrogen electrode (RHE) using the equation E (vs. RHE) = E (vs. Hg/HgO) + E$^0$ (Hg/HgO) + 0.059*pH in the 1.0 M KOH. LSV curves were collected with a scan rate of 5 mV s$^{-1}$ and the LSV curves shown in the manuscript were corrected by IR compensation (E$_{iR-corrected}$ = E$_{original}$ – I * R$_s$). The chron-oamperometry curve was performed for the durability test at the potential of −0.1 V vs. RHE. Electrochemical surface area (ECSA) was obtained by conducting cyclic voltammetry (CV) in the range of 0.1–0.2 V vs. RHE with various scan rates from 10 to 50 mV s$^{-1}$. The current density differences (Δj = j$_a$ − j$_c$) were plotted against scan rates, and the slopes can be used to derive C$_{dl}$ and the ECSA. EIS measure-ments were carried out at the potential of −0.1 V (vs. RHE) in the frequency range of 100,000–0.001 Hz with a perturbation of 5 mV. TOF values were calculated using a previously reported method, in which the number of active sites was estimated as the amount of surface sites (including C, Mo, and S atoms)[11]. The

Faradic efficiency of C–MoS$_2$ was evaluated in a H-type cell with an anion exchange membrane as the separator and 20 mL 1.0 M KOH as the electrolyte in each compartment, with a gas chromatography (HA GC-9560) for the hydrogen gas detection. The gas production was detected by the gas chromatography and the Faradic efficiency was calculated using the formula: Faradic efficiency = 2 F × N$_{H_2}$/Q = 2 F × N$_{H_2}$/(It), where F is the Faradic constant, I is the current, t is the running time and N$_{H_2}$ is the amount of H$_2$ production.

**Density functional theory calculation**. All the DFT calculations were carried out using the CASTEP program in Material Studio package of Accelrys Inc[51]. The exchange-correlation functional was employed by the Perdew-Burke-Ernzerhof (PBE) of generalized gradient approximation (GGA) with the ultrasoft pseudopo-tentials (USP). The van der Walls interactions was considered by the DFT disper-sion correction (DFT-D). The number of plane wave was determined by an energy cutoff of 450 eV. The Brillouin zone was sampled by a 2 × 2 × 1 k-points grid for the structure optimizations and double k-points meshes were employed for the density of states (DOS) calculations, respectively. A complete linear synchronous transitions (LST) and quadratic synchronous transitions (QST) approach was used for the transition state searching. The convergence tolerances were set to 2.0 * 10$^{-6}$ eV per atom for energy, 0.002 Å for maximum displacement, and 0.05 eV Å$^{-1}$ for maximum force. The surfaces were modeled by a periodic slab repeated in 6 * √2 * 2 √2 surface unit cell with a vacuum region of 13 Å between the slabs along the z axis. The H$_2$O absorption energy was calculated using the following equation, ΔE$_{H_2O}$ = E$_{surf−H_2O}$ − E$_{surf}$ − E$_{H_2O}$, where E$_{surf−H_2O}$ and E$_{surf}$ are the total energies of the surface covered with and without H$_2$O molecule, E$_{H_2O}$ is the energy of free H$_2$O molecule.

## Data availability

The authors declare that the main data to support the finds of this study are available within the article and its Supplementary Information. Extra data are available from the corresponding author upon request

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

## Acknowledgements
We thank the financial support of the National Key Research and Development Program of China (2017YFA0206703), Natural Science Fund of China (No. 21771169, 11722543, 11505187), the Fundamental Research Funds for the Central Universities (WK2060190074, WK2060190081, and WK2310000066), USTC start-up funding, and Recruitment Program of Global Expert. We also acknowledge the beamline BL10B at National Synchrotron Radiation Laboratory, and the Beamline BL14W1 at Shanghai Synchrotron Radiation Facility for XPS and XAFS characterizations. The numerical calculations in this paper have been done on the supercomputing system in the Super-computing Center of University of Science and Technology of China.

## Author contributions
G.W. and Y.Q. designed and supervised the project. Y.Z. and S.N. conducted the project and contributed equally to this work. J.Y., Y.W., X.Z. and J.Zhu conducted XPS and XAS measurements. J.C., Y.X., Y.L. and J.Z. performed SEM and TEM characterization. X.L. and S.N. provided the DFT calculation. G.W., Y.Z. and X.L. wrote and revised the manuscript, and all the authors discussed the results and commented on the manuscript.

## Additional information

**Competing interests:** the authors declare no competing interests.

