## [Peer Review File · Nature Communications]

Reviewers' comments:

Reviewer #1 (Remarks to the Author):

In this paper, the authors devised a way to synthesise MoS₂ using Mo₂C precursor in thermal sulphur treatment. The authors claimed C doping can be achieved through "incomplete" sulphurisation of Mo₂C, which lead to decent HER performance in alkaline condition. As stated in the introduction, HER in alkaline is preferred because of the durability and low cost of the electrolyser setup.

In my assessment, the HER performance of the reported catalyst is very good in alkaline, although more thorough analysis of exchange current density comparison with Pt should be included. I find the HER performance of Mo based compounds are getting difficult to track as quite a number of new results are being published every week. I note that MoP/doped graphene also shows similar HER performance in alkaline with η below 50 mV for 10 mA/cm² (10.1021/acscatal.7b00555), which dampens the novelty slightly.

The main novelty of this work is the proposed hybrid Mo-C-S structure in "incomplete" sulphurisation of Mo₂C. However at this current version, the evidence is not very convincing that the authors have obtained such structure. My reasoning is elaborated below:

Determination of C doping using XPS is actually not very straightforward, because now the energy alignment cannot be done using adventitious C1s spectra anymore. I do not think the authors did a thorough job in analysing the XPS data, the C carbides should be in the range of 282 – 283.5 eV (10.1016/j.apsusc.2015.11.089, 10.1021/acsaem.7b00054), while it is taken at the adventitious C1s position at Figure S4. Now this calls into question authors' understanding about their system. I also note the S2s position in main Figure 2a are shifted with respect to MoS₂, which is not expected. I also note that the MoS₂ reference is also synthesised from Mo₂C (means the presence of C cannot be excluded). As the authors would like to highlight their Mo-C-S sample, properly synthesised MoS₂ with minimum interference from C should be used instead. I also note that the procedure for XPS data treatment and fitting is absent, thus it is not possible to gauge the soundness of the authors' approach.

The use of EXAFS to investigate possibilities of C doping into MoS₂ lattice is appropriate. I also appreciate that the authors conducted numerical fitting to the EXAFS data. However, the data treatment fitting procedure is also absent in the current version, so I cannot evaluate the authors' method fairly. The basis of the proposed C doping is the RDF plot (Fig 2c and 2d). However, I think this evidence is thin, because the position marked with Mo-C could very well be Mo-O (10.1021/ja510328m, 10.1051/jp4:19972136). Different EXAFS fitting models with C and O in the first shell should be compared, some difference should be observable as C and O X-ray scattering factor differs by a factor of 4 at the energy range around 20 keV. If O contribution (e.g. by XPS) is examined, I'm sure the authors would see Mo-O contribution.

I'm not sure how the DFT models proposed (Figure S11) reflects the real condition in the sample. In Figure S11, the Mo-C distance is probably less than half that of Mo-S (the actual structural information used in the DFT is absent, I can't tell the actual distances). However, in the EXAFS fitting data, Mo-C is only about 0.3 Å shorter than Mo-S. I am also not sure how the author account for the charge balance in the system, as to accommodate 1 C atom, some S vacancy is required if formal oxidation of Mo(IV) is expected.

At the current state, I do not think the authors could not provide convincing evidence of C doping into the MoS₂ structure, and thus the theoretical assessment is not very useful (and potentially misleading)

as it is not reflective of the actual sample condition. For this reason I would not recommend publication in Nature Communication.

Reviewer #2 (Remarks to the Author):

In comparison with acidic HER, alkaline HER is more sluggish due to its larger energy barrier. Generally, MoS₂ is not highly active for alkaline HER. In this work, the authors demonstrate the carbon-doped MoS₂ as a high-performance HER electrocatalysts in alkaline solutions. However, the current structural characterizations can not solidly prove the carbon doping in the MoS₂. Meanwhile, the utilization of Ag/AgCl reference electrode make the HER activity unreliable. Therefore, I can not recommend its publication in Nature Communications.

1. In Figure S2, for C-MoS₂, some weak peaks are not distinguished, which originate from the Mo₂C. In addition, from the XRD pattern, XANES and XPS analyses, the Mo₂C is still be observed in C-MoS₂. The C-MoS₂ may be a composite of Mo₂C/MoS₂. The authors should give more solid proofs for C-doped.
2. How about the loading weight of C-MoS₂ on the carbon cloth?
3. For the HER tests in alkaline solutions, Ag/AgCl reference is not stable so that achieved HER performance is not reliable. Hg/HgO electrode can be used.
4. During the chronoamperometric measurements, what is the potential?
5. What is the carbon content in the C-MoS₂? How about the influence of carbon content on the HER performance of C-MoS₂?
6. The Faradic efficiency should be provided.
7. The HER performance should be also compared with state-of-the-art HER electrocatalysts like Mo₂C and NiMo alloy.
8. The reaction pathway on Mo₂C need to be evaluated as a comparison.

Reviewer #3 (Remarks to the Author):

This is a careful and comprehensive study on a new class of C-MoS₂, who shows excellent HER performance in alkaline condition. They showed that by introducing C into the MoS₂ lattice, the molecular orbitals and its electronic state were significantly altered. Complementary studies show that this modification promotes both H₂O adsorption and dissociate, leading to enhanced HER performance. The claims are supported by series of careful characterizations and corresponding electrochemical measurements. In light of the performance, it is useful to put things in perspective of a bigger picture. It is useful for the authors to compare the performances among the state of art HER in alkaline, inclusive of all recent advancements from both precious metal and non-precious metal based catalyst. I recommend the publication of the manuscript in Nature communication after minor revisions.

Point-by-point response to the referees' comments

We sincerely thank the referees for carefully reviewing our manuscript and their valuable comments, which certainly help improve our manuscript. We also appreciate the opportunity the editor has given us, to address the comments and revise the manuscript. The changes in the revised manuscript have been highlighted in yellow for your review. The point-by-point responses are presented below.

Reviewers' comments:

Reviewer #1 (Remarks to the Author):

In this paper, the authors devised a way to synthesise MoS₂ using Mo₂C precursor in thermal sulphur treatment. The authors claimed C doping can be achieved through "incomplete" sulphurisation of Mo₂C, which lead to decent HER performance in alkaline condition. As stated in the introduction, HER in alkaline is preferred because of the durability and low cost of the electrolyser setup.

Response: We thank the referee for carefully reviewing our manuscript. Aiming at the naturally inert HER activity of MoS₂ in alkaline condition, we rationally introduced carbon dopants in MoS₂ through a partial sulfurization of Mo₂C and successfully endowed MoS₂ with exceptional alkaline HER catalysis via carbon-induced orbital modulation. Although it is well recognized that MoS₂ is inactive for alkaline HER, the underlying essence is still unclear. In this work, our work indicates the surface orbital orientation of the MoS₂ are unfavorable for water adsorption and activation due to the large steric effect, and the designed C doping strategy can well tune the orbital orientation to boost the alkaline water dissociation kinetics. This work is conceptually novel and could provide a new vision for the construction of MoS₂-based HER catalysts and beyond.

In my assessment, the HER performance of the reported catalyst is very good in alkaline, although more thorough analysis of exchange current density comparison with Pt should be included.

Response: We sincerely thank the referee for the positive comments on the catalyst performance, and we also warmly appreciate the valuable suggestion. Following the suggestion, we have carefully analyzed the exchange current densities in the revised manuscript. The exchange current density basically reflected the intrinsic electron transfer rate between electrode and electrolyte, which could provide insights on the intrinsic catalytic activities of catalysts. The exchange current densities for Mo₂C, MoS₂, C-MoS₂ and Pt/C derived by Tafel extrapolation are 0.12, 0.25, 0.87 and 1.28 mA cm⁻², respectively. Clearly, although the exchange current density of C-MoS₂ is a

little smaller than that of Pt/C, it is substantially larger than that of MoS₂, suggesting carbon doping can change the intrinsic catalytic activities of MoS₂ for alkaline HER catalysis. Meanwhile, we also estimated the turnover frequencies (TOF) of the C-MoS₂ and MoS₂ (Figure 3d, in the manuscript) for HER catalysis. The substantially larger TOF values of C-MoS₂ than that of MoS₂ also indicate carbon doping can essentially promote the per-site activity of MoS₂ for HER catalysis. We have given corresponding discussion on it in the revised manuscript.

I find the HER performance of Mo based compounds are getting difficult to track as quite a number of new results are being published every week. I note that MoP/doped graphene also shows similar HER performance in alkaline with η below 50 mV for 10 mA/cm² (10.1021/acscatal.7b00555), which dampens the novelty slightly.

Response: We thank the referee's comments on the novelty of our work and we are pleased to clarify this issue. We agree that quite a number of Mo-based compounds are being published every week. However, evaluating the impact/novelty of a work should reply on what it has done to solve the remaining scientific or engineering challenges/questions of the studied materials, not only on the number of published papers. As we have mentioned above, it is well recognized that MoS₂ is naturally inert for alkaline HER catalysis; however, the underlying essence is still unclear. Aiming at this issue, our orbital analysis indicates the surface orbital orientation of the MoS₂ is unfavorable for water adsorption and activation, due to the large steric effect. Moreover, our rationally designed C dopant in MoS₂ can uniquely vacate an unhybridized 2p_z orbital perpendicular to the basal surface, which can facilitate the water adsorption and association and thus endow MoS₂ with exceptional alkaline HER activity. The idea to employ orbital tuning for the alkaline HER catalysis of MoS₂ is conceptually novel.

For the reference paper mentioned by the referee (10.1021/acscatal.7b00555), they synthesized MoP decorated on S, N dual-doped graphene for HER and displayed 49 mV at 10 mA cm⁻² in 1.0 M KOH solution. First, the materials are different and the scientific questions faced by each material are also different. Moreover, the reference work did not mention any orbital tuning effects of MoP or S, N dual-doped graphene on the alkaline HER catalysis, also suggesting the targeted research motivations and scientific questions are different. Second, the HER performance of our C-MoS₂ represents the best alkaline HER activity among the ever-reported MoS₂, and it is also better than that of MoP reported in the reference paper. Taken together, we reasonably believe our work is conceptually novel and the existence of MoP paper could not affect the novelty of our work.

The main novelty of this work is the proposed hybrid Mo-C-S structure in "incomplete" sulphurisation of Mo₂C. However at this current version, the evidence is not very convincing that the authors have obtained such structure. My reasoning is elaborated below:

Response: Actually, the main novelty of our work is to decipher the modulation essence of orbital orientation on the alkaline HER catalysis of MoS₂. We also sincerely appreciate the constructive and valuable suggestions raised by the referee, which certainly help to improve our manuscript. Following the suggestions, we have carefully conducted more experiments to clarify the concerns of the referee.

Determination of C doping using XPS is actually not very straightforward, because now the energy alignment cannot be done using adventitious C1s spectra anymore. I do not think the authors did a thorough job in analysing the XPS data, the C carbides should be in the range of 282 – 283.5 eV (10.1016/j.apsusc.2015.11.089, 10.1021/acsaem.7b00054), while it is taken at the adventitious C1s position at Figure S4. Now this calls into question authors' understanding about their system. I also note the S2s position in main Figure 2a are shifted with respect to MoS₂, which is not expected.

Response: We thank the referee for the valuable comments, which is very important to verify the C doping. We agree that using C1s for the energy alignment may not be accurate to determine C doping, which could result in errors in the binding energy due to the adventitious C 1s position. After discussing this issue with the XPS technicians, we further employed Au 4f as the external calibration reference for the energy alignment by sputtering tiny amounts of Au particles, and recollected the XPS spectra for all the samples, as shown in Figure R1. Considering the C1s calibration may result in errors in the energy alignments, we thus refitted the newly acquired XPS spectra with Au 4f as calibration reference. In comparison with Mo 3d spectrum (Figure R1a) of MoS₂, the Mo 3d doublets of C-MoS₂ have obvious shoulders at the lower binding energy region, clearly suggesting Mo has more than one chemical state. Given that the C-MoS₂ is obtained by partially sulfurizing Mo₂C, the chemical states for the broad Mo 3d spectrum can be probably assigned to Mo-S and Mo-C. Meanwhile, the fitted C-Mo position in the newly obtained C 1s spectrum (Figure R1b) is located at around 282.9 eV, which is also well consistent to the reasonable range. The fitting parameters are displayed in Table R1 and R2. In addition, the slight shift of S 2s position suggests the chemical environments around S atoms slightly change after carbon doping, which could originate from the electronegativity difference between C and S. When C is doped into MoS₂ with C-Mo-S state, the C with less electronegativity and electron pulling ability will result in more electron density around the nearby S sites, and thus exhibits a slightly negative shift in binding energy of S 2s. All the XPS analysis and data fitting have been well gone through by the professional XPS technician. We have updated the newly obtained XPS results in the revised manuscript as Figure 2a and Figure S4, and given corresponding discussion on them.

Figure R1. The XPS Mo 3d (a) and C 1s (b) spectra of Mo₂C, MoS₂ and C-MoS₂, respectively.

Table R1. The fitting parameters of the XPS Mo 3d spectra.

Samples	Binding energy		Assignment	FWHM	
	3d _{5/2}	(3d _{3/2})		3d _{5/2}	(3d _{3/2})
C-MoS ₂	229.3	(232.4)	S-Mo-S	1.06	(1.09)
	228.4	(231.5)	S-Mo-C	0.94	(0.96)
MoS ₂	229.5	(232.6)	S-Mo-S	1.09	(1.13)
Mo ₂ C	228.0	(231.1)	C-Mo-C	0.97	(1.02)

Table R2. The fitting parameters for the XPS C 1s spectra.

Samples	Binding energy	Assignment	FWHM
C-MoS ₂	283.0	C-Mo	1.66
	284.5	C-C	1.64
	286.0	C-O and C=O	1.77
MoS ₂	284.5	C-C	1.68
	285.9	C-O	1.79
Mo ₂ C	282.9	C-Mo	1.65
	284.5	C-C	1.67
	286.0	C-O and C=O	1.78

I also note that the MoS_2 reference is also synthesised from Mo_2C (means the presence of C cannot be excluded). As the authors would like to highlight their Mo-C-S sample, properly synthesised MoS_2 with minimum interference from C should be used instead.

Response: We thank the referee for the insightful comments. We agree that C may not be completely excluded by sulfurization at high temperature. However, based on the XPS Mo 3d spectra in Figure R1a, the Mo 3d peaks of the prepared MoS_2 at the high temperature of 900 °C are more symmetrical than those of C- MoS_2 , suggesting the Mo-C states have been well converted to Mo-S at the high temperature and the existence of Mo-C in the MoS_2 is very limited. In addition, following the suggestions, we also fabricated MoS_2 with minimum carbon interference, by thermal sulfurization of ammonium molybdate with S powder as the sulfur source (*J. Am. Chem. Soc.* 2016, 138, 16632-16638). Figure R2a-c show the XRD, SEM and HRTEM of the newly prepared MoS_2 sample, suggesting we have successfully prepared MoS_2 . We also studied the electrochemical properties of the newly prepared MoS_2 and compared the performance with C- MoS_2 in Figure R2d. Obviously, the performance of MoS_2 with minimum carbon interference is much lower than that of C- MoS_2 , clearly revealing carbon dopants play substantial roles on the HER activity enhancement. We have added the control samples in the revised supporting information as Figure S13 and given corresponding discussion on it.

Figure R2. (a) XRD pattern, (b) SEM and (c) HRTEM images of MoS_2 synthesized by thermal sulfurization of ammonium molybdate with S powder as the sulfur source. (d) The LSV curves of C- MoS_2 and the MoS_2 for alkaline HER catalysis.

I also note that the procedure for XPS data treatment and fitting is absent, thus it is not possible to gauge the soundness of the authors' approach.

Response: We thank the referee for the comments and we are pleased to provide more details on the data treatment and fitting. The obtained binding energies were first calibrated using the Au 4f energy (Au 4f_{7/2}, binding energy of 84 eV). The XPS fitting was performed using the XPS PEAK41 software with Shirley background subtraction. For the XPS fitting, we cannot treat it just as a black-box tool, which will always yield inaccurate results. Actually, understanding the chemistry is more important to conduct XPS fitting, as it suggests the possible number of chemical states. [Practical surface analysis, 2nd edition, Vol. 1, Auger and X-ray photoelectron spectroscopy. D. Briggs & M. P. Seah] Given that the C-MoS₂ is prepared by the conversion from Mo₂C via controlled sulfurization and the obtained Mo 3d spectrum for C-MoS₂ displays obvious shoulders at lower binding energy region, it suggests at least two chemical states (S-Mo-S and S-Mo-C) exist in the C-MoS₂. Similarly, C 1s spectrum of C-MoS₂ also exhibits a negative shift of binding energy in comparison with MoS₂, suggesting the possible existence of Mo-C states. The XPS database (NIST X-ray Photoelectron Spectroscopy Database, <https://srdata.nist.gov/xps/intro.aspx>) was used as a reference to assign the possible chemical states. During fitting, the peak intensity ratio and the full width at half maximum (FWHM) were well constrained in a reasonable range. The fitting parameters are summarized in Table R1 and R2. All the fitting details have been gone through by the XPS technician in our structural analysis center. We have added the data fitting details in the Experimental Section of the revised manuscript.

Table R1. The fitting parameters of the XPS Mo 3d spectra.

Samples	Binding energy	Assignment	FWHM
	3d _{5/2} (3d _{3/2})		3d _{5/2} (3d _{3/2})
C-MoS ₂	229.3 (232.4)	S-Mo-S	1.06 (1.09)
	228.4 (231.5)	S-Mo-C	0.94 (0.96)
MoS ₂	229.5 (232.6)	S-Mo-S	1.09 (1.13)
Mo ₂ C	228.0 (231.1)	C-Mo-C	0.97 (1.02)

Table R2. The fitting parameters for the XPS C 1s spectra.

Samples	Binding energy	Assignment	FWHM
C-MoS ₂	283.0	C-Mo	1.66
	284.5	C-C	1.64
	286.0	C-O and C=O	1.77

MoS ₂	284.5	C-C	1.68
	285.9	C-O	1.79
Mo ₂ C	282.9	C-Mo	1.65
	284.5	C-C	1.67
	286.0	C-O and C=O	1.78

The use of EXAFS to investigate possibilities of C doping into MoS₂ lattice is appropriate. I also appreciate that the authors conducted numerical fitting to the EXAFS data. However, the data treatment fitting procedure is also absent in the current version, so I cannot evaluate the authors' method fairly.

Response: We appreciate the referee's positive comments on our efforts and we are also very pleased to provide more details in the EXAFS data processing. The EXAFS data were disposed according to the standard procedures through the ATHENA module implemented in the IFEFFIT software packages. The quantitative curve-fittings were performed for the Fourier transformed k^2 -weighted $\chi(k)$ in the R-space with a Fourier transform k-space range of 3.1-11 Å⁻¹ by employing the module ARTEMIS 4 of IFEFFIT. In addition, the backscattering amplitude F(k) and phase shift $\Phi(k)$ were acquired by FEFF8.0 code. For the curve-fitting, all the amplitude reduction factor S_0^2 was set to the best-fit value of 0.85 determined from fitting the data of MoS₂. To fit the curve in the R-range of 1.1-3.4 Å, we considered Mo-C, Mo-Mo and Mo-S paths as the central-peripheral. For each path, the structural parameters, like coordination number (N), interatomic distance (R) and edge-energy shift (ΔE_0) were opened to be varied. For the Mo-C, Mo-S and Mo-Mo coordination, the common adjustable parameters of ΔE_0 and σ^2 were employed to reduce the number of free parameters. The structural fitting parameters are summarized in Table S4 and the fitting curves of C-MoS₂ and MoS₂ are plotted in Figure 2e and Figure S7, respectively. All the data processing was performed under the guidance of the EXAFS technicians in Shanghai Synchrotron Radiation Facility, China. We have added the fitting details in the Experimental Section of the revised manuscript.

The basis of the proposed C doping is the RDF plot (Fig 2c and 2d). However, I think this evidence is thin, because the position marked with Mo-C could very well be Mo-O (10.1021/ja510328m, 10.1051/jp4:19972136). Different EXAFS fitting models with C and O in the first shell should be compared, some difference should be observable as C and O X-ray scattering factor differs by a factor of 4 at the energy range around 20 keV. If O contribution (e.g. by XPS) is examined, I'm sure the authors would see Mo-O contribution.

Response: We sincerely thank the referee for the constructive comments and we are pleased to clarify this issue. We totally agree with the referee on the fact that the

positions of Mo-C and Mo-O are very close and it is difficult to technically separate them. Actually, for the assignment, we cannot conduct the data analysis just as a black box. It could be more meaningful and convincing to assign the chemical bond states in the RDF plots, by combining the chemical states of the studied materials and other surface analysis techniques. Our reasons for the Mo-C assignment are as follows.

First, oxygen signal was indeed observed in the XPS survey spectrum, although its intensity is substantially lower than the carbon. The observed oxygen signal probably originates from the surface C-O residuals (XPS C1s in Figure R1b) after the carbonization of the polymeric precursors during synthesis, while the Mo-O signal is substantially suppressed in the XPS Mo 3d spectra of Mo₂C, C-MoS₂ and MoS₂ (Figure R1a), clearly suggesting the surface Mo-O state is very limited. In addition, XPS, a typical surface sensitive technique, only gives the surface chemical information, while EXAFS provides bulk averaging information. Therefore, if we cannot observe obvious Mo-O states in the XPS studies, it most probably won't be observed in the EXAFS spectra. Meanwhile, considering C-MoS₂ is prepared by partially sulfurizing Mo₂C, it is thus most likely to assign the peaks at ~1.3 Å in the RDF plot as the Mo-C states.

Second, following the referee's suggestion, we also performed the EXAFS fitting using Mo-C and Mo-O paths. Figure R3a and 3b show the fitting curves for the FT Mo k-edge curve of C-MoS₂ with different fitting models, and Table R3 displays the corresponding fitting parameters. The obtained R_f value for the Mo-O model (2.9%) is slightly larger than the R_f of the Mo-C path (1.1%), also suggesting the Mo-C path is more likely than the Mo-O path in the C-MoS₂. Furthermore, we also performed more-refined wavelet transform (WT) with high resolution in both k and R spaces of Mo K-edge EXAFS oscillation analysis for the control samples of MoO₂ and Mo₂C, as shown in Figure R3c. Clearly, the MoO₂ possesses a maximum intensity at k = 6 Å⁻¹, while Mo₂C owns a contour centered at 4.5 Å⁻¹, which is very close to the contour centered at 4.6 Å⁻¹ in C-MoS₂. This result also suggests it is more likely to assign the peak at ~1.3 Å as the Mo-C than the Mo-O.

Besides probing the chemical states of Mo-O by the spectroscopic approaches, we also studied the roles of surface oxidation on the catalytic properties of MoS₂ and indirectly excluded the assignment for Mo-O. We intentionally introduce surface oxidation on the MoS₂ (prepared with minimum carbon interference, *J. Am. Chem. Soc.* 2016, 138, 16632-16638) by annealing MoS₂ in air condition at various temperatures. Figure R4a shows the corresponding polarization curves for HER catalysis. Clearly, after the controlled surface oxidation, the HER performance does not show significant enhancement and the performance even decays at the higher oxidation temperature. In addition, the overall performances of MoS₂ and surface oxidized MoS₂ are substantially lower than that of C-MoS₂, suggesting surface oxidation is not the main enhancement factor for the exceptional alkaline HER activity. Furthermore, theoretical analysis also reveals that oxygen incorporation in MoS₂ cannot well tune the orbital orientation, which is totally different from the designed carbon doping. Taken together, we reasonably believe it is most likely to assign the peaks at ~1.3 Å in the RDF plot as the Mo-C. We also added a brief

discussion on the assignment of Mo-C in the revised manuscript.

Figure R3. Radial distribution of Fourier-transformed EXAFS signal of C-MoS₂ with a Mo-O path (a) and Mo-C (b) at R=1.3 Å. The whole contour plots of the wavelet transform (WT) of MoO₂ (c) and Mo₂C (d).

Figure R4. (a) The LSV curves of the MoS₂ and the MoS₂ with thermal oxidation in air at various conditions. (b) The top-view (upper) and side-view (lower) orbital compositions of the conduction band in O doped MoS₂.

Table R3. The EXAFS fitting parameters for C-MoS₂ with Mo-C and Mo-O path.

	Path	R (Å)	N	σ^2 (Å ²)	ΔE_0 (eV)	R _f (%)
C-MoS₂	Mo-C	2.10	1.2	0.0027	2.28	1.1
	Mo-S	2.38	4.9	0.0023		
C-MoS₂	Mo-O	2.06	1.0	0.0031	1.13	2.9
	Mo-S	2.39	5.2	0.0029		

I'm not sure how the DFT models proposed (Figure S11) reflects the real condition in the sample. In Figure S11, the Mo-C distance is probably less than half that of Mo-S (the actual structural information used in the DFT is absent, I can't tell the actual distances). However, in the EXAFS fitting data, Mo-C is only about 0.3 Å shorter than Mo-S.

Response: We appreciate the valuable comments and we are pleased to clarify this issue. The illusion of the bond length is probably due to the side-view angle, which leads to the Mo-C bond length looks much less than half of the Mo-S bond in the direction perpendicular to the basal plane. Actually, either the Mo-C bond or Mo-S bond is not perpendicular to the basal plane and the calculated average Mo-C distance is ~2.03 Å in our DFT model, which corresponds well with the Mo-C distance of ~2.10 Å based on the EXAFS fitting results. We have provided the detailed structural information of MoS₂ and C-MoS₂ used for calculation in Figure R5.

Figure R5. The top (upper) and side (lower) views of MoS₂ (a) and C-MoS₂ (b) with detailed Mo-S and Mo-C bond lengths, respectively.

I am also not sure how the author account for the charge balance in the system, as to accommodate 1 C atom, some S vacancy is required if formal oxidation of Mo(IV) is expected.

Response: We agree that sulfur vacancies could be probably created after carbon doping and we warmly appreciate the referee's constructive comments, which really help to understand the catalytic mechanism of C-MoS₂. Following the suggestion, we have modified our theoretical models by introducing sulfur vacancies. Meanwhile, we also experimentally studied the role of sulfur vacancies on the alkaline HER catalysis. The vacancy concentration used in the model is based on the charge balance, while the Mo-C/Mo-S ratio is obtained by the XPS Mo 3d spectra. In order to equally distribute the positions of the doped C and the generated S vacancies, we considered seven different configurations and calculated their corresponding HER catalytic pathways, respectively. All the possible structures of C-MoS₂ shown in Figure R6 consistently generate unhybridized 2p_z orbitals perpendicular to the basal plane on the C sites, which could provide water adsorption and activation sites, and meanwhile suggest the existence of S vacancies doesn't have substantial effects on the orbital orientation of the C sites. Moreover, the simulated catalytic pathways in Figure R7 further indicate the HER reactions on C-MoS₂ proceed with much lower energy barriers than that on the MoS₂, revealing the orbital tuning induced by carbon doping could substantially boost the alkaline HER activity. We have incorporated the updated results in the revised manuscript as Figure 4 and Figure S14-S21, and also given corresponding discussion on them.

Furthermore, we also provided the theoretical analysis of the MoS₂ with only sulfur vacancies in Figure R8, which indicates that sulfur vacancy doesn't have obvious effects on the orbital orientation tuning for water adsorption and activation. The corresponding water adsorption energy on the MoS₂-Vs is -0.23 eV, which is only slightly higher than that on the MoS₂. Meanwhile, we experimentally studied the role of sulfur vacancies on the HER catalysis. We annealed the MoS₂ under hydrogen environment at different temperatures to create sulfur vacancies, which is a common approach to create sulfur vacancies in MoS₂ (*Nano Lett.* 2016, 16, 4047-4053). Figure R8 shows the electrochemical properties of the hydrogen treated MoS₂ (H-MoS₂) under different hydrogen treatments. As we can see, the alkaline HER performance slightly increases, after hydrogen treatment. However, the overall performance of the H-MoS₂ is still far less than that of C-MoS₂, suggesting the sulfur vacancy is not be the main factor for the exceptional alkaline HER catalysis of the C-MoS₂. Taken together, the incorporation of sulfur vacancies in the C-MoS₂ could make our model more reasonable. Even so, the main enhancement factor for the exceptional alkaline HER catalysis is probably still stemmed from the carbon-induced orbital tuning. We have added a brief discussion on the role of vacancies in the revised manuscript.

Figure R6. The top views (upper) of MoS₂ and C-MoS₂ structures, and the corresponding side view (lower) of empty orbital orientations with different distributions of carbon dopants and the involved vacancies.

Figure R7. The relative energy diagrams along the reaction coordinate, including the first (left panel) and second (right panel) water dissociation process on the basal plane of MoS₂ and C-MoS₂ with the corresponding seven distribution of carbon dopants and vacancies, respectively. R: Reactant, RC: Reactant Complex, TS: Transition State, IM: Intermediate.

Figure R8. (a) The top and side views of the empty orbitals close to the Fermi level in MoS₂ with only vacancies. The red circles denote the vacancy positions. (b) The LSV curves of MoS₂ synthesized on carbon cloth by thermal sulfurization of ammonium molybdate and the H₂ treated counterparts at different temperatures.

At the current state, I do not think the authors could not provide convincing evidence of C doping into the MoS₂ structure, and thus the theoretical assessment is not very useful (and potentially misleading) as it is not reflective of the actual sample condition. For this reason I would not recommend publication in Nature Communication.

Response: Finally, we sincerely express our gratitude to the referee for all the constructive comments and suggestions, which really help improve our understanding on the relation between structures and properties of C-MoS₂. Following the referee's suggestions, we have conducted more experimental and theoretical analysis to provide more convincing evidences of C doping and correspondingly modify the theoretical models to reflect the most possible sample condition. Although it is still a great challenge for theoretical calculation to provide a complete description for the real and complicated experimental system, especially for the surface electrocatalysis, the theoretical analysis based on simplified model system still can help understand the surface reaction mechanism (Nørskov J. K. et al. Density functional theory in surface chemistry and catalysis, *PNAS*, 2011, 108, 937-943). We believe the revised version of our manuscript should be acceptable for Nature Communications.

Reviewer #2 (Remarks to the Author):

In comparison with acidic HER, alkaline HER is more sluggish due to its larger energy barrier. Generally, MoS₂ is not highly active for alkaline HER. In this work, the authors demonstrate the carbon-doped MoS₂ as a high-performance HER electrocatalysts in alkaline solutions. However, the current structural characterizations can not solidly prove the carbon doping in the MoS₂. Meanwhile, the utilization of

Ag/AgCl reference electrode make the HER activity unreliable. Therefore, I can not recommend its publication in Nature Communications.

Response: We thank the referee for carefully reviewing our manuscript and the raised valuable comments. For the structural analysis, we have provided more detailed and convincing evidences to prove the C doping in the C-MoS₂. Meanwhile, we also used Hg/HgO as the reference electrode to replace the previous Ag/AgCl reference and remeasured all the electrochemical characterization of the samples. Our point-by-point response is presented as follows.

1. In Figure S2, for C-MoS₂, some weak peaks are not distinguished, which originate from the Mo₂C. In addition, from the XRD pattern, XANES and XPS analyses, the Mo₂C is still be observed in C-MoS₂. The C-MoS₂ may be a composite of Mo₂C/MoS₂. The authors should give more solid proofs for C-doped.

Response: We thank the reference for the constructive comments and we are pleased to clarify this issue. In Figure S2, we have magnified the region, which shows the undistinguished peaks originate from the noise. Meanwhile, our XRD, XPS and XANES studies also did not show any obvious existence of Mo₂C in the C-MoS₂. XRD patterns indicate almost all the diffraction peaks of Mo₂C disappear after sulfurization, where the peaks at the ~39.2 and 60.2 degree are located at the same position as MoS₂. Moreover, EXAFS basically provide the bulk averaging structural information, which can give more refined and convincing structural information than XRD. Figure R9a shows the Mo k-edge extended XAFS oscillation function $k^2\chi(k)$. Clearly, the oscillation of C-MoS₂ is very similar to MoS₂, but quite different from Mo₂C, convincingly indicating the C-MoS₂ possesses similar structure to MoS₂ and there is no obvious existence of Mo₂C in C-MoS₂. Moreover, Figure 2c in the main text is the Fourier transform (FT) of Mo K-edge of the studied samples. It mainly gives the information on the distances between central and neighboring atoms, which can be used to determine the existence of Mo-C bond states, but cannot be used to determine the existence of Mo₂C, because C doped MoS₂ also owns Mo-C bond states. More importantly, XPS Mo 3d spectrum in Figure 2a also clearly shows the C-Mo-C chemical state in Mo₂C positively shifts to higher binding energy, indicating the S-Mo-C state in C-MoS₂ has been created without obvious C-Mo-C chemical state remaining. Moreover, even after deep Ar ion profiling for 10 min under 3 kV, prominent sulfur signal can still be detected (shown in Figure R9b), indicating the inner part of the C-MoS₂ is still MoS₂, not Mo₂C. Taken together, all these results consistently support there is no Mo₂C in the prepared C-MoS₂. We have added a brief discussion on it in the revised manuscript.

Figure R9. (a) Mo k-edge extended XAFS oscillation function $k^2\chi(k)$ of MoS₂, C-MoS₂ and Mo₂C. (b) XPS S 2p spectra of C-MoS₂ before and after Ar ion profiling.

2. How about the loading weight of C-MoS₂ on the carbon cloth?

Response: The loading weight of C-MoS₂ on the carbon cloth is about 1.8 mg cm⁻². We have added the loading weights of the catalysts in the experimental section.

3. For the HER tests in alkaline solutions, Ag/AgCl reference is not stable so that achieved HER performance is not reliable. Hg/HgO electrode can be used.

Response: We thank the referee for the valuable suggestion, which can help us more reliably evaluate the catalytic performance. Following the suggestion, we have employed Hg/HgO electrode as the reference and remeasured the electrochemical performance, as shown in Figure 3 in the revised manuscript. With the Hg/HgO reference electrode, the C-MoS₂ still can deliver an impressive overpotential of 45 mV at 10 mA cm⁻². We have updated these data in the revised manuscript.

4. During the chronoamperometric measurements, what is the potential?

Response: For the chronoamperometric measurement, the applied potential is -0.1 V vs. RHE without IR correction. We have added the details in the experimental section.

5. What is the carbon content in the C-MoS₂? How about the influence of carbon content on the HER performance of C-MoS₂?

Response: We appreciate the referee's constructive comments. The carbon content in C-MoS₂ is estimated to be ~16.7 at.% (atomic ratio of C-Mo/(S-Mo+C-Mo)) based on the area of Mo-C and Mo-S states in the XPS Mo 3d spectrum of C-MoS₂. We also studied the effects of carbon contents on the HER performance by controlling the

sulfurization temperatures. Figure R10 shows the HER activities with different carbon contents. With the increase of carbon concentration, the catalytic activity of C-MoS₂ increases and reaches a maximum value. However, further increasing the C ratio will result in performance decay, probably due to too much destruction of the layered structures of MoS₂. We have added a brief discussion on it in the revised manuscript.

Figure R10. The carbon content dependent HER activities.

6. The Faradic efficiency should be provided.

Response: The Faradic efficiency of C-MoS₂ was evaluated in a H-type cell with an anion exchange membrane as the separator and 20 mL 1.0 M KOH as the electrolyte in each compartment, with a gas chromatography (HA GC-9560) for the hydrogen gas detection. The HER Faradic efficiency of C-MoS₂ was estimated to be ~97%. We have added the measured Faradic efficiency and the measurement details in the revised manuscript.

7. The HER performance should be also compared with state-of-the-art HER electrocatalysts like Mo₂C and NiMo alloy.

Response: Following suggestion, we have made two tables for the performance comparison. Table R4 is the performance comparison among the ever-reported MoS₂-based catalysts in alkaline condition. Table R5 shows the comparison between C-MoS₂ with other typical HER catalysts in alkaline condition including precious metals and no-precious catalysts such as Mo₂C, NiMo and Pt. Although the performance of C-MoS₂ is not the best one among the HER catalysts including non-precious and precious catalysts, the C-MoS₂ obviously displays the best performance among the ever-reported MoS₂-based catalysts. Moreover, the targeted scientific question of this work is to decipher the modulation essence of orbital orientation on the alkaline HER catalysis of MoS₂. Although pursuing high-performance catalysts is important for catalysis research, fundamentally understanding the catalytic essence is also always equally important, as it could

provide valuable insights for the catalyst design in future. We have added the comparison tables in the revised manuscript and correspondingly given brief discussion on it.

Table R4. The HER performance comparison of C-MoS₂ with the ever-reported MoS₂-based catalysts in alkaline condition.

Catalysts	Overpotentials at 10 mA cm ⁻² (mV)	Tafel Slopes (mV dec ⁻¹)	Electrolyte	Ref.
C-MoS₂	45	46	1.0 M KOH	This work
CoMoS _x	> 220	-	PH=13	Nat. Mater. 15 , 197-203 (2016)
MoS ₂ /NiCo-LDH on CFP	78	76.6	1.0 M KOH	Joule 1 , 383-393 (2017)
3D macroporous MoS ₂ film/Mo foil	184	87	1.0 M KOH	Electrochim. Acta 168 , 133-138 (2015)
Ni(OH) ₂ /MoS ₂	80	60	1.0 M KOH	Nano Energy 37 , 74-80 (2017)
MoS ₂ /Ni ₃ S ₂	110	83.1	1.0 M KOH	Angew. Chem. Int. Ed. 55 , 6702-6707 (2016)
Co-MoS ₂ /BCCF-21	48	85	1.0 M KOH	Adv. Mater. 30 , 1801450 (2018)
N, Mn-MoS ₂	66	50	1.0 M KOH	ACS Catal. 8 , 7585-7592 (2018)
Ni doped MoS ₂ nanosheets on CC	98	60	1.0 M KOH	Energy Environ. Sci. 9 , 2789-2793 (2016)
MoS ₂ @Ni/CC	91	89	1.0 M KOH	ACS Appl. Mater. Interfaces 8 , 14521-14526 (2016)
Mo-N/C@MoS ₂	117	64.3	1.0 M KOH	Adv. Funct. Mater. 27 , 1702300 (2017)

Table R5. HER performance comparison with the ever-reported other non-precious and precious metal-based catalysts in alkaline condition.

Catalysts	Overpotentials at 10 mA cm ⁻² (mV)	Tafel Slopes (mV dec ⁻¹)	Electrolyte	Ref.
C-MoS₂	45	46	1.0 M KOH	This work
Mo ₂ C/NCF	100	65	1.0 M KOH	ACS Nano 10 , 11337-11343 (2016)
Mo ₂ C-GNR	121	54	0.1 M KOH	ACS Nano 11 , 384-394 (2017)

MoO _{3-x} nanosheets on CC	140	56	0.1 M KOH	Adv. Energy Mater. 6 , 1600528 (2016)
Mo ₂ C/CC	140	124	1.0 M KOH	J. Mater. Chem. A 3 , 16320–16326 (2015)
Mo _x C	82	-	0.1 M KOH	ACS Nano 25 , 7527–7533 (2017)
N,P-doped Mo ₂ C@C	47	71	1.0 M KOH	ACS Nano 10 , 8851–8860 (2016)
Nanospheres Mo ₂ C@NC	60	-	1.0 M KOH	Angew. Chem. Int. Ed. 54 , 10752–10757 (2015)
MoNi ₄	15	30	1.0 M KOH	Nat. Commun. 8 , 15437 (2017)
Mo ₂ C@2D-NPC	45	46	1.0 M KOH	ACS Nano 11 , 3933–3942 (2017)
Mo ₂ C@NPC	72	52	1.0 M KOH	(2017)
MoP ₂ NS/CC	67	70	1.0 M KOH	J. Mater. Chem. A 4 , 7169–7173 (2016)
Pt ₃ Ni ₃ NWs/C	40	-	1.0 M KOH	Angew. Chem. Int. Ed. 55 , 12859–12863 (2016)
Ru–MoO ₂	29	31	1.0 M KOH	J. Mater. Chem. A 5 , 5475–5485 (2017)
Pt ₁₃ Cu ₇₃ Ni ₁₄	148 (η5)	54	1.0 M KOH	ACS Appl. Mater. Interfaces 8 , 3464–3472 (2016)
MoO _x /Ni ₃ S ₂ /NF	106	90	1.0 M KOH	Adv. Funct. Mater. 26 , 4839–4847 (2016)
NF–Ni ₃ Se ₂ /Ni	203	79	1.0 M KOH	Nano Energy 24 , 103–110 (2016)

8. The reaction pathway on Mo₂C need to be evaluated as a comparison.

Response: We sincerely thank the referee for the valuable suggestion. We have performed the catalytic pathway on Mo₂C, as shown in Figure R11. Obviously, the reaction on the Mo₂C proceeds on a potential energy surface with much higher energy barrier than that on C-MoS₂. Moreover, the generated hydrogen adsorbed intermediate (Sur-H) is so stable that the adsorbed hydrogen needs overcome 1.37 eV in energy for H desorption from the carbon site of the surface, which potentially impedes the HER catalysis. The related results have been added as Figure S23 and also been given a brief discussion on it in the revised manuscript.

Figure R11. The relative energy diagram along the reaction coordinate, including the first (left panel) and second (right panel) water dissociation process on the Mo₂C (001), respectively. R: Reactant, RC: Reactant Complex, TS: Transition State, IM: Intermediate.

Reviewer #3 (Remarks to the Author):

This is a careful and comprehensive study on a new class of C-MoS₂, who shows excellent HER performance in alkaline condition. They showed that by introducing C into the MoS₂ lattice, the molecular orbitals and its electronic state were significantly altered. Complementary studies show that this modification promotes both H₂O adsorption and dissociate, leading to enhanced HER performance. The claims are supported by series of careful characterizations and corresponding electrochemical measurements. In light of the performance, it is useful to put things in perspective of a bigger picture. It is useful for the authors to compare the performances among the state of art HER in alkaline, inclusive of all recent advancements from both precious metal and non-precious metal based catalyst. I recommend the publication of the manuscript in Nature communication after minor revisions.

Response: We sincerely appreciate the positive comments on our work and the recommendation for publication after minor revision. Following the suggestions, we have made two tables for the performance comparison. Table R4 is the performance comparison among the ever-reported MoS₂-based catalysts in alkaline condition. Table R5 shows the comparison between C-MoS₂ with other typical HER catalysts in alkaline condition including precious and no-precious catalysts. Although the performance of C-MoS₂ is not the best one among the HER catalysts including non-precious and precious metal-based catalysts, the C-MoS₂ obviously displays the best performance among the ever-reported MoS₂-based catalysts. Moreover, the

targeted scientific question of this work is to decipher the modulation essence of orbital orientation on the alkaline HER catalysis of MoS₂. Although pursuing high-performance catalysts is important for catalysis research, fundamentally understanding the catalytic essence is also always equally important, as it could provide valuable insights for the catalyst design in future. We have added the comparison tables in the revised manuscript and correspondingly given brief discussion on it.

Table R4. HER performance of the ever-reported MoS₂-based catalysts in alkaline condition.

Catalysts	Overpotentials at 10 mA cm ⁻² (mV)	Tafel Slopes (mV dec ⁻¹)	Electrolyte	Ref.
C-MoS₂	45	46	1.0 M KOH	This work
CoMoS _x	> 220	-	PH=13	Nat. Mater. 15 , 197-203 (2016)
MoS ₂ /NiCo-LDH on CFP	78	76.6	1.0 M KOH	Joule 1 , 383-393 (2017)
3D macroporous MoS ₂ film/Mo foil	184	87	1.0 M KOH	Electrochim. Acta 168 , 133-138 (2015)
Ni(OH) ₂ /MoS ₂	80	60	1.0 M KOH	Nano Energy 37 , 74-80 (2017)
MoS ₂ /Ni ₃ S ₂	110	83.1	1.0 M KOH	Angew. Chem. Int. Ed. 55 , 6702-6707 (2016)
Co-MoS ₂ /BCCF-21	48	85	1.0 M KOH	Adv. Mater. 30 , 1801450 (2018)
N, Mn-MoS ₂	66	50	1.0 M KOH	ACS Catal. 8 , 7585-7592 (2018)
Ni doped MoS ₂ nanosheets on CC	98	60	1.0 M KOH	Energy Environ. Sci. 9 , 2789-2793 (2016)
MoS ₂ @Ni/CC	91	89	1.0 M KOH	ACS Appl. Mater. Interfaces 8 , 14521-14526 (2016)
Mo-N/C@MoS ₂	117	64.3	1.0 M KOH	Adv. Funct. Mater. 27 , 1702300 (2017)

Table R5. HER performance of the ever-reported non-precious metal and precious metal-based catalysts in alkaline condition.

Catalysts	Overpotentials at 10 mA cm ⁻² (mV)	Tafel Slopes (mV dec ⁻¹)	Electrolyte	Ref.
-----------	---	--	-------------	------

C-MoS₂	45	46	1.0 M KOH	This work
Mo ₂ C/NCF	100	65	1.0 M KOH	ACS Nano 10 , 11337–11343 (2016)
Mo ₂ C–GNR	121	54	0.1 M KOH	ACS Nano 11 , 384–394 (2017)
MoO _{3-x} nanosheets on CC	140	56	0.1 M KOH	Adv. Energy Mater. 6 , 1600528 (2016)
Mo ₂ C/CC	140	124	1.0 M KOH	J. Mater. Chem. A 3 , 16320–16326 (2015)
Mo _x C	82	-	0.1 M KOH	ACS Nano 25 , 7527–7533 (2017)
N,P-doped Mo ₂ C@C Nanospheres	47	71	1.0 M KOH	ACS Nano 10 , 8851–8860 (2016)
Mo ₂ C@NC	60	-	1.0 M KOH	Angew. Chem. Int. Ed. 54 , 10752–10757 (2015)
MoNi ₄	15	30	1.0 M KOH	Nat. Commun. 8 , 15437 (2017)
Mo ₂ C@2D-NPC	45	46	1.0 M KOH	ACS Nano 11 , 3933–3942 (2017)
Mo ₂ C@NPC	72	52	1.0 M KOH	(2017)
MoP ₂ NS/CC	67	70	1.0 M KOH	J. Mater. Chem. A 4 , 7169–7173 (2016)
Pt ₃ Ni ₃ NWs/C	40	-	1.0 M KOH	Angew. Chem. Int. Ed. 55 , 12859–12863 (2016)
Ru–MoO ₂	29	31	1.0 M KOH	J. Mater. Chem. A 5 , 5475–5485 (2017)
Pt ₁₃ Cu ₇₃ Ni ₁₄	148 (η5)	54	1.0 M KOH	ACS Appl. Mater. Interfaces 8 , 3464–3472 (2016)
MoO _x /Ni ₃ S ₂ /NF	106	90	1.0 M KOH	Adv. Funct. Mater. 26 , 4839–4847 (2016)
NF–Ni ₃ Se ₂ /Ni	203	79	1.0 M KOH	Nano Energy 24 , 103–110 (2016)

Reviewers' comments:

Reviewer #1 (Remarks to the Author):

Comments have been addressed.

Reviewer #2 (Remarks to the Author):

I appreciate that the authors have basically addressed my comments. However, there are still several comments that need to be carefully considered.

- 1) How to exclude the contributions of increased surface area and conductivity of C-MoS₂ to its HER activity?
- 2) 20-h stability test is not enough.
- 3) Based on current results, it is still difficult to confirm the catalyst composition, carbon-doped MoS₂ or MoS₂/Mo₂C nanocluster hybrids. First, the XRD peaks of C-MoS₂ and Mo₂C are seriously overlapped. Second, the strong interfacial interactions between MoS₂ and Mo₂C nanoclusters will also cause strong variations of Mo-S, Mo-C and Mo-Mo bonds. Specially, the C-MoS₂ is prepared by sulfurization of Mo₂C. Is it a homogeneous sulfurization process or surface-to-bulk sulfurization process?
- 4) Although MoS₂ does not show the state-of-the-art HER performance, it is an ideal catalyst model for revealing and engineering the active sites. Here, the HER activity of C-MoS₂ is much lower than reported alkaline HER catalysts (Mo₂C, NiMo alloys).
- 5) After HER stability test, the chemical and bonding information of C-MoS₂ are not provided.

Point-by-point response to the referees' comments

We sincerely thank the referees for carefully reviewing our manuscript and their valuable comments. We also appreciate the opportunity given to us, to further address the comments. The changes in the revised manuscript have been highlighted in yellow for your review. The point-by-point responses are presented below.

Reviewers' comments:

Reviewer #1 (Remarks to the Author):

Comments have been addressed.

Response: We sincerely appreciate the comments raised by the referee, which really help improve our manuscript. We are also very pleased that the referee is satisfied with our previous response and revision.

Reviewer #2 (Remarks to the Author):

I appreciate that the authors have basically addressed my comments. However, there are still several comments that need to be carefully considered.

Response: We are very pleased to know that our previous response can basically address the referee's comments. We also have carefully considered the new and valuable comments raised by the referee, and prepared a point-by-point response to these comments.

1) How to exclude the contributions of increased surface area and conductivity of C-MoS₂ to its HER activity?

Response: We thank the referee for the valuable comments. For the surface area contribution, we actually have studied the effects of surface area variation on the catalytic performance in the previous version. We estimated the electrochemical surface area variation by employing the electrochemical double layer capacitance (Figure S10). To further eliminate the influence of surface area and probe the intrinsic per-site activity, turnover frequency (TOF), a surface area independent figure of merit, was also calculated in Figure R1 (Figure 3d, in manuscript). All these results clearly reveal that carbon doping can essentially promote the intrinsic catalytic activity of MoS₂ for alkaline HER.

For the conductivity contribution, it is basically impossible to decouple the two parameters of the C doping and the conductivity, due to the intrinsic correlation between carbon doping induced-electronic structures and electrical properties. Although electrical conductivity is important for catalysis, it is not the only parameter to evaluate the catalytic essence. For example, graphene and metals (Ti, Cu, Al *et al.*) own excellent electrical conductivity, but their catalytic HER performance is

intrinsically poor. Even for 1T MoS₂ with better conductivity, it still does not exhibit such exceptional alkaline HER catalysis in literature [ACS Catal.2014, 4, 3957-3971]. Therefore, we reasonably believe that the excellent alkaline HER activity of C-MoS₂ is mainly attributed to the C-induced orbital tuning for water adsorption and dissociation. We have provided a brief discussion on the conductivity and surface area issues in the revised manuscript.

Figure R1. The potential-dependent TOF curves of Mo₂C, MoS₂ and C-MoS₂.

2) 20-h stability test is not enough.

Response: We thank the referee for the valuable suggestion. Following the suggestion, we have performed a longer durability test for 240 h, as shown in Figure R2. Clearly, it does not show obvious current decay, suggesting C-MoS₂ possesses superior catalytic stability for alkaline HER electrolysis. We have updated the durability data in Figure 3f.

Figure R2. Chronoamperometric curve of C-MoS₂ recorded at -0.1 V vs. RHE without IR correction for a continuous 240 h.

3) Based on current results, it is still difficult to confirm the catalyst composition, carbon-doped MoS₂ or MoS₂/Mo₂C nanocluster hybrids. First, the XRD peaks of C-MoS₂ and Mo₂C are seriously overlapped. Second, the strong interfacial

interactions between MoS₂ and Mo₂C nanoclusters will also cause strong variations of Mo-S, Mo-C and Mo-Mo bonds. Specially, the C-MoS₂ is prepared by sulfurization of Mo₂C. Is it a homogeneous sulfurization process or surface-to-bulk sulfurization process?

Response: We appreciate the referee's comments and we are pleased to clarify this issue. First, we agree that there are two XRD peaks of MoS₂ (C-MoS₂) and Mo₂C are overlapped. If we suppose the material is the MoS₂/Mo₂C composite, the strong diffraction intensity of the overlapped peak may suggest the ratio of Mo₂C is not low and the crystal domain size (~8.1 nm) should be also easily detectable based on the Debye-Scherrer equation. However, our HRTEM images (shown in Figure R3a) did not find obvious Mo₂C domains. In addition, X-ray absorption fine structure spectroscopy (XAFS) can give more refined structural information than XRD. We further collected the Mo K-edge extended XAFS oscillation functions $k^2\chi(k)$ of the MoS₂ mixed with small amounts of Mo₂C (denoted as Mo₂C + MoS₂), and compared the spectrum with those of MoS₂, Mo₂C and C-MoS₂, as shown in Figure R3b. Apparently, C-MoS₂ owns very similar oscillation profiles with that of MoS₂, indicating they have similar crystal structures. However, for the Mo₂C + MoS₂, there are several distinct features at $k = 7.2, 8.7$ and 9.7 \AA^{-1} , which has been highlighted by the dash circles. The quite different oscillation profiles between C-MoS₂ and the sample of Mo₂C + MoS₂, clearly indicate these two samples have different scattering properties, further revealing there is no obvious existence of Mo₂C in the prepared C-MoS₂.

Second, the interfacial interaction could cause the variation of Mo-S, Mo-C and Mo-Mo bonds. Actually, such effects only work on the interfacial several atom layers, while the structural information probed by the XAFS is the average value of the bulk phase. Since the interfacial atom number is far less than the bulk atom numbers, the effects of interfacial interaction on the average radial distances of the atom-atom vectors should be very limited. Moreover, the XPS Mo 3d (shown in Figure R3c) profiles in C-MoS₂ exhibit obvious shoulders at the low binding energy region, suggesting the existence of more than one chemical states of Mo. The Mo 3d spectrum of C-MoS₂ can be further deconvoluted into two pairs of peaks corresponding to the chemical states of S-Mo-S and S-Mo-C, which is quite different from the C-Mo-C states in Mo₂C and further suggests the C is doped into MoS₂.

For the sulfurization, we think it is probably a surface-to-bulk process, which can be supported by the XRD patterns at various sulfurization temperatures (Figure S9a). At the temperature below 500 °C, Mo₂C cannot be completely converted to MoS₂. We also conducted the depth-profiling XPS analysis on the C-MoS₂. Even after deep Ar ion profiling for 10 min under 3 kV, prominent sulfur signal still can be detected (shown in Figure R3d), indicating the inner part of the C-MoS₂ still undergoes sulfurization. Moreover, the catalytic activity of the sample (shown in Figure S8) with incomplete sulfurization (MoS₂/Mo₂C) is also far less than the performance of C-MoS₂, further revealing C doping probably plays vital role in the HER catalysis. On the other hand, surface-to-bulk sulfurization may lead to inhomogeneous dopant distribution. Actually, in the theoretical calculations, we also considered the effects of

the C dopant distribution on the water dissociation (Figure S15). All the possible dopant distribution of C dopants in the C-MoS₂ consistently generate unhybridized 2pz orbitals perpendicular to the basal plane on the C sites, which could provide water adsorption and activation sites for HER catalysis. Taken together, our results clearly indicate there is no obvious existence of Mo₂C in the prepared C-MoS₂. We have added a brief discussion on this issue in the revised manuscript and also discussed the possibility of Mo₂C cluster in C-MoS₂ that is beyond the detection limits of the currently used characterization techniques

Figure R3. (a) The HRTEM image of C-MoS₂. (b) The Mo k-edge extended XAFS oscillation functions $k^2\chi(k)$ of MoS₂, C-MoS₂, Mo₂C and the physical mixture of Mo₂C and MoS₂. (c) The XPS core-level Mo 3d spectra of Mo₂C, MoS₂, and C-MoS₂, respectively. (d) The XPS S 2p spectra of C-MoS₂ before and after Ar ion profiling.

4) Although MoS₂ does not show the state-of-the-art HER performance, it is an ideal catalyst model for revealing and engineering the active sites. Here, the HER activity of C-MoS₂ is much lower than reported alkaline HER catalysts (Mo₂C, NiMo alloys).

Response: We sincerely thank the referee for realizing our work could provide an ideal catalyst model to reveal and engineer the active sites. For the catalytic performance, the C-MoS₂ is actually better than the pure Mo₂C without further doping treatment, but lower than the reported NiMo alloys. Typically, different catalysts may face different scientific challenges for catalytic application, due to their own intrinsic electronic structures. In this work, the targeted scientific question is to decipher the modulation essence of MoS₂ for the alkaline HER catalysis and propose a possible resolution strategy to address this issue by tuning the orbital orientation. Although the performance of C-MoS₂ is not the best one among all kinds of the HER catalysts including non-precious and precious catalysts, the C-MoS₂ displays the best

performance among the ever-reported MoS₂-based catalysts. While pursuing high-performance is important for catalysis research, fundamentally understanding the catalytic essence is also always equally important, as it could provide valuable insights for the catalyst design in future. We have briefly discussed this issue in the revised manuscript.

5) After HER stability test, the chemical and bonding information of C-MoS₂ are not provided.

Response: We warmly appreciate the referee's constructive suggestion. Following the suggestion, we have conducted XPS and XAFS to probe the chemical and bonding information after the durability test, as shown in Figure R4. Clearly, there is no big difference in chemical and bonding states of the C-MoS₂ before/after the stability test, suggesting the robustness of the C-MoS₂ for HER catalysis. We have added these data as Figure S14 in the supporting information and given a brief discussion in the revised main text.

Figure R4. (a) The normalized Mo K-edge XANES spectra. (b) The Fourier transform (FT) of Mo K-edge. XPS core-level Mo 3d (c) and C 1s (d) spectra of the C-MoS₂ before and after the durability test.

REVIEWERS' COMMENTS:

Reviewer #2 (Remarks to the Author):

My comments have been well addressed. I recommend its publication without revisions.

Point-by-point response to the referees' comments

Reviewers' comments:

Reviewer #2 (Remarks to the Author):

My comments have been well addressed. I recommend its publication without revisions.

Response: We sincerely appreciate the comments raised by the referee, which really help improve our manuscript. We are also very pleased to know the referee is satisfied with our previous response and revision.